# FedEL: Federated Elastic Learning for Heterogeneous Devices

**Letian Zhang**
Middle Tennessee State University
Murfreesboro, TN 37132
letian.zhang@mtsu.edu

**Bo Chen**
Middle Tennessee State University
Murfreesboro, TN 37132
bc7b@mtmail.mtsu.edu

**Jieming Bian**
University of Florida
Gainesville, FL 32611
jieming.bian@ufl.edu

**Lei Wang**
University of Florida
Gainesville, FL 32611
leiwang1@ufl.edu

**Jie Xu**
University of Florida
Gainesville, FL 32611
jie.xu@ufl.edu

## Abstract

Federated learning (FL) enables distributed devices to collaboratively train machine learning models while maintaining data privacy. However, the heterogeneous hardware capabilities of devices often result in significant training delays, as straggler clients with limited resources prolong the aggregation process. Existing solutions such as client selection, asynchronous FL, and partial training partially address these challenges but encounter issues such as reduced accuracy, stale updates, and compromised model performance due to inconsistent training contributions. To overcome these limitations, we propose FedEL, a federated elastic learning framework that enhances training efficiency while maintaining model accuracy. FedEL introduces a novel window-based training process, sliding the window to locate the training part of the model and dynamically selecting important tensors for training within a coordinated runtime budget. This approach ensures progressive and balanced training across all clients, including stragglers. Additionally, FedEL employs a tensor importance adjustment module, harmonizing local and global tensor importance to mitigate biases caused by data heterogeneity. The experiment results show that FedEL achieves up to 3.87× improvement in time-to-accuracy compared to baselines while maintaining or exceeding final test accuracy.

## 1 Introduction

Federated learning (FL) is a privacy-preserving machine learning paradigm where distributed clients, such as mobile devices and IoT systems, collaboratively train a global model while keeping their data local. Typically, FL involves devices performing local model training and sharing parameters with a central server for global model updates. However, heterogeneous hardware capabilities among devices lead to "straggler", or slower clients, causing significant training delays as the server must wait for their updates. This challenge hinders the scalability of FL, particularly in large-scale cross-device scenarios.

**Status Quo and Their Limitations.** To address computational constraints, existing solutions fall into three main categories: client selection, asynchronous FL, and partial training. *Client Selection* (Figure 1, top-left). Selecting a subset of devices for training based on specific criteria can mitigate delays. However, significant differences in clients' data distributions often leave stragglers underrepresented, reducing the global model's accuracy. *Asynchronous FL* (Figure 1, top-right). This approach decouples local training from global aggregation, allowing stragglers to train independently. While this

39th Conference on Neural Information Processing Systems (NeurIPS 2025).

reduces delays, the global model often relies on faster clients, leaving stragglers' contributions infrequent and potentially outdated, which may harm convergence [44]. *Partial Training* (Figure 1 bottom left). Techniques like width and depth scaling adjust the model architecture to accommodate varying resources. Width scaling resizes convolutional layers, risking channel mismatches during aggregation [18], while depth scaling limits training to early layers, leading to suboptimal task-specific features and reduced performance [42]. These limitations highlight the need for a novel training paradigm to overcome resource heterogeneity and enable high-performance FL in real-world deployments.

ElasticTrainer [14] introduces a method for selecting important deep neural network (DNN) tensors to meet runtime training requirements on *a single device*. By focusing on these key tensors, ElasticTrainer accelerates training. When applied to FL, it offers a potential solution for addressing stragglers by allowing each client to select important tensors based on its hardware capabilities under a unified runtime constraint. This ensures that all clients complete local training within a similar timeframe, enabling synchronized global model aggregation. However, directly deploying Elastic-Trainer in FL scenarios has two limitations: *Limitation#1 Limited Training Scope on Slower Clients:* Due to the chained rule of DNN backward propagation, unselected tensors still compute gradients to propagate updates to the selected tensors. This constrains the selected tensors on slower devices to the back-end of the DNN, reduc-

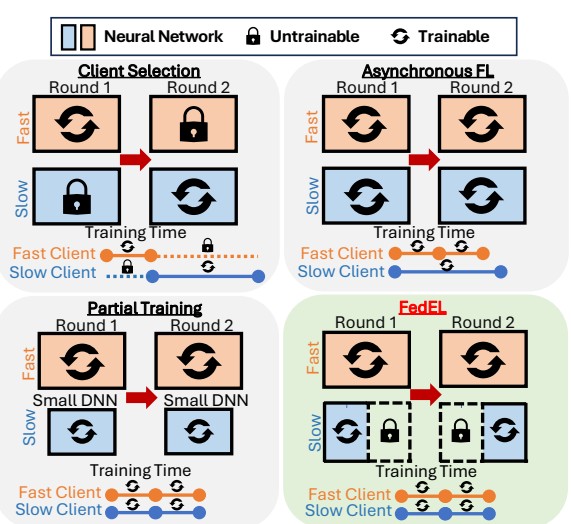

Figure 1: Existing works vs. FedEL.

ing training on the front-end feature extraction layers and degrading FL accuracy, especially with heterogeneous data distributions. *Limitation#2 Exacerbated Local Model Drift:* Variations in data distribution cause significant differences in tensor importance across clients. Training only the important tensors amplifies local model drift, where client models diverge from the global model, further reducing accuracy.

**Overview of the Proposed Approach.** Motivated by the above limitations, we propose FedEL, a federated elastic learning framework that enhances federated training efficiency. To address the first limitation, we propose a window-based training approach that divides the DNN model into multiple blocks, ensuring that each part of the model is trained during FL rounds. Before training, we use a tensor timing profiler to measure the training time for each tensor, which is then aggregated into block-level training times. In each FL round, the window slides to include a set of blocks based on the runtime budget and current training status. The sliding window process involves moving the front edge to include deeper blocks and shrinking the end edge to exclude blocks that no longer require training. ElasticTrainer is then modified to select important training tensors within the selected window, allowing straggler clients to progressively train the crucial tensors of the entire DNN model. To address the second limitation, we design a tensor importance adjustment module. At the start of each FL round, the client estimates the global model's tensor importance using the global models from the current and previous rounds, along with the learning rate. This global tensor importance is used to adjust the local tensor importance computed by ElasticTrainer, ensuring tensor selection considers both local and global data distributions.

**Evaluation.** We implement FedEL on both a hardware testbed and software simulations. The hardware testbed consists of ten NVIDIA Jetson devices connected wirelessly to a server. To simulate large-scale scenarios, we extend the setup with a diverse client simulation. We evaluate FedEL using various DNN models and four real-world FL datasets across three key tasks: image classification, voice command recognition, and next-word prediction. Our results show that: (1) FedEL outperforms baselines on the time-to-accuracy. Specifically, FedEL outperforms FedAvg by $3.87\times$ in time-to-accuracy while final test accuracy is on par with or even higher than FedAvg. (2) FedEL reduces memory overhead and energy consumption during training compared to existing methods. (3) Ablation studies confirm the necessity and importance of each key component in FedEL's design.

## 2 Related Work

**On-single-device training.** Leveraging mobile and embedded computing for DNN model training has gained attention [53]. Some studies reduce computation by quantizing or pruning gradient propagation for certain neurons [4, 10, 36, 16]. Others use a two-stage paradigm, where the system prepares the computing graph and generates a training plan before model training [33, 45, 39, 29, 9, 14]. Our work builds on ElasticTrainer [14], which dynamically selects important tensors during training. However, applying single-device methods directly to FL scenarios can be challenging due to heterogeneous systems and data.

**Heterogeneous federated learning.** To address the challenges posed by low-end devices in FL, three main training methodologies have been proposed: (1) client selection, (2) asynchronous FL, and (3) partial training. *Client selection* methods [21, 25, 6, 42] evaluate the utility of each client and select a subset to participate in FL rounds. For example, PyramidFL [25] ranks clients based on utility. However, when slower clients have unique data, they may be infrequently selected, leading to accuracy loss [26]. *Asynchronous FL* methods [50, 52, 30, 28] allow the global model to be updated as soon as local models are received, bypassing slower clients. TimelyFL [50] adjusts workloads based on client resources, increasing participation. However, this may lead to slower convergence and accuracy issues, as model updates may arrive at different times, causing inconsistencies, particularly with heterogeneous devices and data [44]. *Partial training* involves training part of the model by scaling its width or depth [7, 5, 12, 3, 18, 34, 41]. HeteroFL [7] scales convolutional layers to match devices' available training time. Similar methods include Federated Dropout [5] and FjORD [12], but these can disrupt model architecture and degrade performance. DepthFL [18] customizes models based on client resource constraints, but the global model size is limited by the device with the largest memory. Unlike existing methods, FedEL ensures all clients participate in FL rounds, allowing clients with varying speeds to complete training of the full DNN model by sliding their windows.

## 3 Background and Motivation

To help better understand our design of FedEL, we first introduce how the ElasticTrainer can speed up on-device DNN training with a small accuracy loss. Afterwards, we show the issues of using ElasticTrainer directly in the heterogeneous federated learning framework, hence motivating our federated elastic selection of the trainable DNN portion at runtime.

**ElasticTrainer.** The tensor selection problem in ElasticTrainer [14] is formulated as a constrained optimization problem:

$$\max_{\boldsymbol{A}} \boldsymbol{A} \cdot \boldsymbol{I}, \quad s.t. \ T_{fw} + T_{bw}(\boldsymbol{A}) \leq T_{th}. \tag{1}$$

Here, $\boldsymbol{A}$ is a binary mask representing the selected tensors. $\boldsymbol{I}$ is the importance of tensors. $T_{fw}$ is the fixed forward propagation time, independent of tensor selection. $T_{bw}(\boldsymbol{A})$ represents the backward propagation time, which depends on the selected tensors involved in gradient computation. The sum $T_{fw} + T_{bw}(\boldsymbol{A})$ is the estimated training time constrained to a user-defined runtime threshold $T_{th}$, aimed at accelerating training. For example, setting $T_{th}$ to $50\%$ of the full model training time implies reducing the training time to half that of full model training. ElasticTrainer consists of two modules: the tensor timing profiler and the tensor importance evaluator. The tensor timing profiler creates an *offline* tensor-level backward computation time graph, preserving the execution order of all tensors during backward propagation, from the output to the input layer. In the online training phase, at the start of each fixed interval, the tensor importance evaluator evaluates the importance of all tensors $\boldsymbol{I}$. ElasticTrainer then uses dynamic programming to solve the optimization problem (1), freezes unselected tensors, and trains only the selected tensors during each interval.

**Federated Learning with ElasticTrainer.** In FL, diverse hardware capabilities lead to significant variations in local training times across clients. By employing ElasticTrainer with a uniform runtime threshold $T_{th}$ across all clients, it becomes theoretically feasible for all clients to participate in each FL round, ensuring consistent training times regardless of hardware differences. Consider an FL setup with $N$ clients, starting from the same initial model. In each FL round r, the central server distributes its current model to all clients. Each client $n$ trains the model on its local data using ElasticTrainer with the uniform $T_{th}$, then sends its model update $\boldsymbol{w}_{n,r}$ to the server. The server aggregates the updates as $\boldsymbol{w}_{r+1} = \sum_{n=1}^{N} \boldsymbol{c}_n(t) \odot \boldsymbol{w}_{n,r}$, yielding the global model for next round $r + 1$, where $(\boldsymbol{c}_n(t))_k = \frac{(\boldsymbol{A}_n(t))_k}{\sum_{n \in \mathcal{N}} (\boldsymbol{A}_n(t))_k}$ denotes the k-th tensor selection of mask $\boldsymbol{A}_n(t)$ at training

round $r$. The updated global model $\boldsymbol{w}_{r+1}$ is broadcast to all clients for the next round. This process iterates until a predefined maximum number of training rounds is reached.

To validate this approach, we design the following experiment. **System Platform.** We design a FL system with 10 client devices, consisting of 5 NVIDIA Jetson Xavier NX kits (Xavier) [2] and 5 NVIDIA Jetson Orin kits (Orin) [1]. All devices connect to a PC via WiFi, with Orin offering superior computational performance compared to Xavier. **Dataset and Model.** We focus on an image classification task using the CIFAR10 dataset [19] on VGG16 model [35], implemented within the FedAvg framework [31]. ElasticTrainer [14] is used for local training, and the dataset is partitioned non-iid using a Dirichlet distribution ( $\alpha = 0.1$ ) [46]. **Training Setup.**

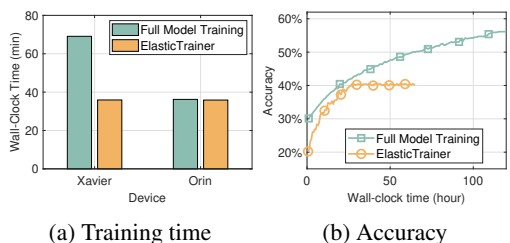

(a) Training time  (b) Accuracy

Figure 2: Average training time per FL round and training accuracy evolution of FedAvg with full model training and FedAvg with ElasticTrainer.

The runtime threshold $T_{th}$ is set based on the full model training time of the faster Orin devices, ensuring all clients complete local training within a similar timeframe.

**Limitations of FL with ElasticTrainer** Figure 2a illustrates the average training time per FL round on Xavier and Orin using FedAvg with full model training and FedAvg with ElasticTrainer. Due to the disparity in computational performance between Xavier and Orin, Xavier's training time per round is nearly twice as long as Orin's when using FedAvg with full model training. Consequently, Orin clients must wait for Xavier clients to complete their training before responding to the central server, leading to longer idle times for the faster Orin clients. Figure 2a

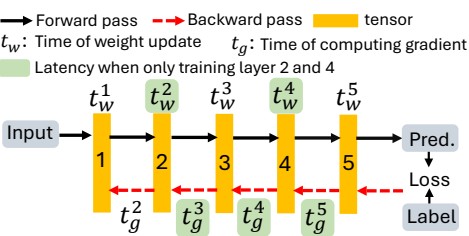

Figure 3: Tensor selection in ElasticTrainer.

also demonstrates that FedAvg with ElasticTrainer reduces this imbalance, enabling both Xavier and Orin to complete each round of training in roughly the same time. However, as shown in Figure 2b, the accuracy of FedAvg with ElasticTrainer is $40.03\%$ lower compared to FL with full model training. In the following sections, we explore in more detail how the direct deployment of ElasticTrainer in FL underutilizes data and training efficiency. These insights are foundational to the design of FedEL.

**Limitation#1: Limited Training Scope on Slower Clients.** ElasticTrainer identifies the most important tensors under a specified training time threshold $T_{th}$. However, the tensor selection process is not straightforward due to the dependencies inherent in backward propagation. Even if a tensor is not selected, it must compute and pass gradients to previous tensors, contributing to the total training time. For example, as illustrated in Figure 3, the backward propagation time comprises two components:

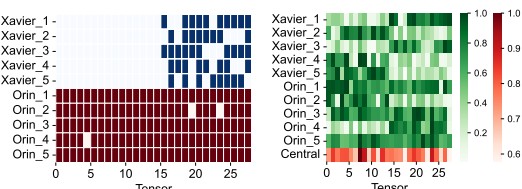

Figure 4: Tensor selection in Xavier's model and Orin's model.

Figure 5: Tensor importance of ten-device FL and central training.

(1) Gradient Computation Time $t_g$: Time spent calculating the gradient of the current tensor to pass to the previous tensor. (2) Weight Update Time $t_w$: Time spent updating the tensor's weights using gradients from the subsequent tensor. If tensors 2 and 4 are selected, the total training time includes both selected and unselected tensors, calculated as $t_g^5 + t_w^4 + t_g^4 + t_g^3 + t_w^2$. ElasticTrainer employs a dynamic programming approach, starting from the last tensor and selecting important tensors in reverse order until the accumulated weight update and gradient computation time reaches $T_{th}$.

Figure 4 demonstrates tensor selection across Xavier and Orin clients during one FL round. While Orin clients (faster devices) can train nearly all tensors[1], Xavier clients (slower devices) tend to focus training on the back part of the DNN model. This leaves the front feature extractor layers largely untrained due to the same $T_{th}$ being applied across all devices. In FL settings with non-iid data, this

---

[1]Unselected tensors on Orin clients result from ElasticTrainer's computational cost optimization process.

imbalance becomes critical. Xavier clients' untrained feature extractor layers fail to adequately learn essential features, weakening the global model's ability to extract meaningful features. Consequently, the overall accuracy of the FL system degrades.

**Limitation#2: Exacerbated Local Model Drift.** ElasticTrainer is optimized for centralized training, where all data resides on a single device. However, FL involves distributed training, and recent studies [37, 48, 49] have highlighted the local model drift challenge arising from non-iid data distribution among clients. Non-iid data can bias tensor importance evaluation, as local models trained on heterogeneous client datasets reflect varying data distributions. Figure 5 compares tensor importance across ten FL clients and centralized training. In FL, tensor importance differs significantly between clients and also the centralized training due to non-iid data. ElasticTrainer's selective training exacerbates this bias by freezing unselected tensors, intensifying local model drift. As a result, when the central server aggregates these biased local models, the global model accuracy suffers compared to full model training in FL.

## 4  FedEL Design

### 4.1  Sliding Window Training

To address Limitation#1, we propose dividing the DNN into multiple blocks and utilizing a window-based scheme that ensures every part of the DNN model has the opportunity to be trained during the FL local training rounds. Specifically, the DNN model is partitioned into $B$ blocks, denoted as $[\theta_1, \theta_2, \ldots, \theta_B]$, based on its original architecture. Each block may consist of one or more layers, preserving the inherent structural integrity of the model. For instance, in VGG16, which follows a chain-like architecture, each layer can be treated as a separate block. In contrast, ResNet50 contains residual structures, so each residual structure can be considered a block, while other layers outside these structures can also be treated as individual blocks.

**Offline Tensor Time Profiling.** Before initiating the online training process, each client uses the tensor timing profiler in ElasticTrainer to measure the training time for each tensor. This offline tensor-level timing data is then aggregated into block-wise training times by summing the training times of all tensors within each block. Assume block $b$ contains a set $K^b$ of tensors. The training time $T^b$ of block b is computed as: $T^b = \sum_{k \in K^b}(t_g^k + t_w^k)$, where $t_g^k$ is the time of computing the gradient, and $t_w^k$ is the time of updating weights for each tensor $k \in K^b$.

**Online Window-Based Training** Using the block-wise training time file, we first initialize the starting window. The initial window begins with the first block, $\theta_1$, and progressively includes subsequent blocks until the cumulative training time just exceeds the user-specified runtime threshold $T_{th}$. Specifically, the initial window consists of $\Theta_0 = \{\theta_1, \ldots, \theta_m\}$, where $\sum_{b \in \{1,\ldots,m-1\}} T^b < T_{th}$ and $\sum_{b \in \{1,\ldots,m\}} T^b \geq T_{th}$. At each FL round, the window slides, and ElasticTrainer is applied to train the corresponding portion of the DNN model. Over time, this approach ensures that the entire model is trained, enabling complete feature extraction from the data. However, as highlighted in Limitation #1, the blocks outside the current window still require time to compute gradients and pass them to the blocks within the

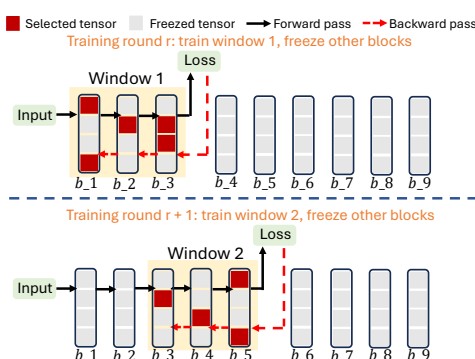

Figure 6: Overview of window-based training in FedEL.

window. This dependency means the original output layer cannot serve as the final output for each window. To address this, a lightweight output layer is attached to the last layer of the window, acting as an early exit. This ensures independent training for each window and facilitates the completion of the window-based training process. **Example.** Figure 6 illustrates the window-based training process with early exits in FL. In round $r$, Window 1, comprising blocks 1, 2, and 3, is selected for training, while the remaining blocks are frozen. The early exit of Window 1 serves as the output layer. Inputs are forwarded through Window 1 to generate predictions, which are used to compute the loss gradient. Backward propagation updates only the weights in Window 1, with other blocks entirely

excluded from forward and backward propagation. This approach applies ElasticTrainer to Window 1, significantly reducing training time. After round $r$, only Window 1's updated weights are sent to the global server for aggregation, and the updated global model is broadcast to all clients for the next round. In round $r + 1$, Window 1 shifts to Window 2, now consisting of blocks 3, 4, and 5. These blocks are trained while others remain frozen. The early exit of Window 2 acts as the output layer. Inputs are forwarded through blocks 1–5 to produce predictions, but only the weights in blocks 3, 4, and 5 are updated during backward propagation. Blocks after block 5 remain frozen, while blocks 1 and 2 participate in forward propagation to pass intermediate results to Window 2. ElasticTrainer continues to optimize training within Window 2.

This iterative and cyclical training process ensures consistent training time across all clients while allowing all parts of the DNN model to be trained, preserving model accuracy.

### 4.1.1 Sliding Window

We assume that the window has two boundaries: the *Front Edge* and the *End Edge*. All blocks between these edges form the current training window. At the beginning of each FL round, clients slide the window to determine the portion of the DNN model to train based on their training progress.

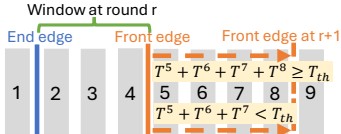 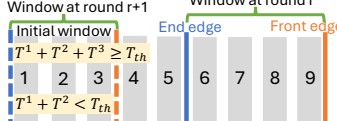 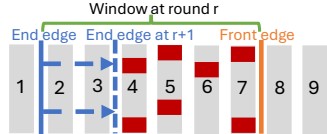

(a) Front edge movement will contain blocks that have accumulated training time just above $T_{th}$.

(b) Front edge movement will revert to the initial window when it touches the end of DNN model.

(c) End edge movement will cull out blocks once there are no important tensors selected from them.

Figure 7: Front edge movement and end edge movement in window sliding.

**Front Edge Movement.** As illustrated in Figure 7a, the front edge moves forward to include deeper blocks of the DNN model. Each movement adds blocks whose cumulative training time slightly exceeds the user-defined runtime threshold $T_{th}$. For instance, in Figure 7a, when training round $r$ begins, the front edge (orange line) shifts to a deeper position (orange dashed line). Here, the cumulative training time of blocks 5, 6, 7, and 8 meets or exceeds $T_{th}$, while the cumulative time for blocks 5, 6, and 7 is below $T_{th}$. If the front edge reaches the end of the DNN model and the cumulative training time of newly added blocks is still below $T_{th}$, this is also considered a front edge movement. Once the front edge reaches the model's end, as shown in Figure 7b, but FL training is not yet complete, the window resets to the initial window for the next round.

**End Edge Movement.** The end edge moves to shrink the training window and freeze blocks that no longer require training. This movement depends on the current training status. If blocks at the window's end are not selected in the previous FL round, the end edge excludes them from the window. This adjustment occurs for two reasons: either the window is too large, preventing ElasticTrainer from selecting important tensors within the threshold $T_{th}$, or ElasticTrainer determines no important tensors exist in those blocks. For example, as shown in Figure 7c, if blocks 2 and 3 contain no important tensors during training round $r$, the end edge will shift to block 4 in the next round.

### 4.1.2 Insert ElasticTrainer into Windows

To integrate ElasticTrainer into windows, we adapt its tensor selection module. In its original form, the module uses dynamic programming to identify the optimal set of important tensors for local training, starting from the last tensor and proceeding until the accumulated training delay, including weight update time and gradient computation time, reaches the runtime threshold $T_{th}$. Our modification adjusts the starting point of dynamic programming to begin at the tensor corresponding to the last layer within the current window. Additionally, we introduce a new base case: if a tensor lies outside the window's range, the dynamic programming process halts and returns the selected important tensors. This adjustment allows ElasticTrainer to be seamlessly applied to window-based training, ensuring efficient and targeted training within each window.

## 4.2 Tensor Importance Adjustment

In Limitation #2, the tensor importance estimated by ElasticTrainer is biased due to the heterogeneous data distribution across clients. To address this bias, we propose a strategy that leverages the global model after aggregation to compute tensor importance. This global tensor importance is then used in the subsequent local training round to adjust the tensor importance at the client side, thereby improving training efficiency. After collecting the locally trained models from all connected clients, the server aggregates these models to next round global model $w_{r+1}$. The aggregated global model is then broadcast back to the clients for the next round of training. ElasticTrainer calculates tensor importance as $\frac{\partial L}{\partial w}\Delta w$, where the loss gradient is multiplied by the tensor update. Upon receiving the updated global model, clients compute the tensor importance of the global model using the formula: $\boldsymbol{I}^g = \frac{w_{r+1}-w_r}{\eta_n} \cdot (w_{r+1} - w_r) = \frac{(w_{r+1}-w_r)^2}{\eta_n}$. Here, $\eta_n$ is the learning rate for client $n$, $\frac{w_{r+1}-w_r}{\eta_n}$ estimates the global model's loss gradient, and $w_{r+1} - w_r$ represents the tensor updates in the global model. The global tensor importance $\boldsymbol{I}^g$ is then used to adjust the local tensor importance for each client as follows: $\boldsymbol{I}_{n,r+1} = \beta \cdot \boldsymbol{I}_{n,r+1} + (1-\beta) \cdot \boldsymbol{I}^g$, where $\beta \in [0,1]$ is a balancing parameter that determines the weighting between local and global importance. This adjustment ensures that local tensor importance aligns better with global priorities, thus improving the overall training accuracy of the model.

Due to page limitations, the complete algorithm and the theoretical convergence analysis of the proposed method are provided in Appendices A and E.

# 5 Evaluation

## 5.1 Experiment Setup

**Datasets, Models, and Tasks.** To demonstrate FedEL's effectiveness across tasks, datasets, and models, we evaluate FedEL on four real-world datasets designed for FL applications at different scales. **Image Classification.** VGG16 [35] model on CIFAR10 dataset [19] and Tiny ImageNet dataset [23]. **Speech Recognition.** ResNet50 [11] model on Google command speech dataset [40]. **Natural Language Processing.** Lightweight Albert [22] model on Reddit dataset [32]. To follow the realistic non-iid data in FL scenarios, we partition the datasets into different clusters using a Dirichlet distribution with $\alpha$ equals 0.1. The Reddit datasets inherently exhibits non-iid characteristics.

**Baselines.** The following baselines are adopted for evaluation purposes: (1) FedAvg [31]. (2) ElasticTrainer [14]. (3) HeteroFL [7]. (4) DepthFL [18]. (5) PyramidFL [25]. (6) TimelyFL [50]. (7) FIARSE [41]. Detailed descriptions of these baseline methods are provided in the Appendix.

**FL Setup.** To evaluate FedEL's effectiveness, we conduct experiments in two scenarios: a small-scale practical edge device setup and a large-scale simulation. *Small-scale Practical Edge Device Scenario:* FedEL is deployed on ten heterogeneous edge devices, comprising five NVIDIA Jetson Xavier NX kits (Xavier) [2] and five NVIDIA Jetson Orin kits (Orin) [1], connected via WiFi to a central PC. Due to the limited number of devices, we evaluate performance using only the CIFAR10 dataset. *Large-scale Simulation Scenario:* To simulate a larger environment, we use tensor timing profiles generated by ElasticTrainer's offline tensor profiler on Orin as a baseline. From this profiling data, we simulate four types of heterogeneous devices with scaled tensor training times, including devices matching the baseline profiling time, devices with $1/2$ of the profiling time, devices with $1/3$ of the profiling time, devices with $1/4$ of the profiling time. A total of 100 clients are simulated, with each randomly assigned a device type and corresponding processing time. This simulation is conducted on a PC equipped with an NVIDIA 3090 GPU. For fair comparisons with baseline methods, unless stated otherwise, the runtime threshold $T_{th}$ is set to the full model training time of the fastest device, and the balance parameter $\beta$ is fixed at 0.6.

## 5.2 End-to-End Performance

**FedEL accelerates training while maintaining high accuracy.** Table 1 summarizes the final accuracy and wall-clock training time of FedEL and its baselines. FedEL consistently outperforms baselines under the same training rounds. Below is a detailed analysis of the results: **FedAvg:** FedEL achieves comparable accuracy to FedAvg, which trains the full model, but reduces wall-clock training time by $1.87 \times -3.87 \times$. This efficiency arises because FedAvg waits for slower clients to complete

Table 1: Comparison of FedEL with baselines on time-to-accuracy.

| Method | Image Classif. | | | | | | Speech Recog. | | | NLP | | |
| | 10 Devices | | | 100 Devices | | | 100 Devices | | | 100 Devices | | |
| | Acc. ↑ | Time | Speedup | Acc. ↑ | Time | Speedup | Acc. ↑ | Time | Speedup | Perp. ↓ | Time | Speedup |
|---|---|---|---|---|---|---|---|---|---|---|---|---|
| FedAvg [31] | 56.13% | 119.8h | N/A | 33.76% | 563.1h | N/A | 58.04% | 709.8h | N/A | 77.48 | 546.4h | N/A |
| ElasticTrainer [14] | 40.03% | 64.8h | 1.84× | 27.65% | 158.6h | 3.55× | 47.96% | 184.3h | 3.84× | 81.02 | 176.2h | 3.10× |
| HeteroFL [7] | 53.44% | 80.1h | 1.49× | 30.56% | 248.2h | 2.26× | 51.47% | 265.9h | 2.66× | 80.11 | 206.1 | 2.65× |
| DepthFL [18] | 54.89% | 77.3h | 1.54× | 34.14% | 198.3h | 2.83× | 54.23% | 207.4h | 3.42× | 78.08 | 212.4h | 2.57× |
| PyramidFL [25] | 56.24% | 115.7h | 1.03× | 34.70% | 497.4h | 1.13× | 58.12% | 587.4h | 1.21× | 77.68 | 418.2h | 1.31× |
| TimelyFL [50] | 53.74% | 66.3h | 1.81× | 33.53% | 198.1h | 2.84× | 56.49% | 193.2h | 3.67× | 80.91 | 177.6h | 3.07× |
| FIARSE [41] | 56.48% | 71.9h | 1.66× | 33.98% | 191.5h | 2.94× | 58.13% | 198.2h | 3.58× | 77.31 | 191.0h | 2.86× |
| FedEL | **56.51%** | **63.8h** | **1.87×** | **34.96%** | **156.8h** | **3.59×** | **58.26%** | **183.3h** | **3.87×** | 77.23 | **174.5h** | 3.13× |

training, whereas FedEL dynamically selects portions of the DNN for slower clients, enabling all clients to complete local training in roughly the same time. **ElasticTrainer:** While ElasticTrainer speeds up training by up to 3.84× compared to FedAvg, it sacrifices over 28.6% accuracy across four datasets. As noted in Section 3, ElasticTrainer's focus on selecting important tensors only from the back of the DNN on slower clients limits global model feature extraction. FedEL addresses this limitation, achieving 1% − 2% faster training time than ElasticTrainer by leveraging window sliding to reduce tensor selection overhead, while maintaining high accuracy. **HeteroFL:** FedEL improves accuracy by 5.7%-14.4% compared to HeteroFL. The uneven scaling of convolutional layers in HeteroFL compromises parameter training and degrades the model's architecture [18]. Furthermore, HeteroFL requires complex global aggregation for mismatched parameters, increasing training time. **DepthFL:** DepthFL partitions models into sub-models for slower clients and uses self-distillation for knowledge transfer. However, its slower training and reliance on training only the front layers of the DNN for slower clients weaken the global model's ability to learn from their data. FedEL outperforms DepthFL with up to 7.1% higher accuracy. **PyramaidFL:** PyramaidFL synchronizes fast and slow clients by allowing fast clients to train for more epochs, accelerating convergence but not reducing total training time. FedEL achieves 1%-2% higher accuracy than PyramaidFL by ensuring balanced participation of slower clients. **TimelyFL:** FedEL achieves up to 5% higher accuracy compared to TimelyFL. The heterogeneity-aware asynchronous approach in TimelyFL reduces participation rates for slower clients, leading to accuracy loss in heterogeneous data environments. FedEL, by contrast, ensures balanced participation across clients, preserving accuracy. **FIARSE:** FIARSE does not account for the dependency of backward gradient propagation. Specifically, its output layer is fixed as the last layer of the network structure. This results in the unselected tensors in FIARSE need to compute and propagate gradients to previously selected tensors.

**FedEL reduces the memory and energy consumption.** Figures 8 and 9 compare FedEL with baselines in terms of memory usage, power consumption, and energy consumption, as measured using the Jetson Power GUI on Xavier and Orin devices. Since the differences in measurements between the two devices are negligible, we present the averaged results to save space. As shown in Figure 8, FedEL reduces memory usage by up to 32.7% compared to FedAvg. This improvement stems from train-

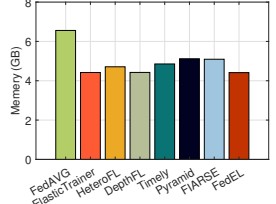
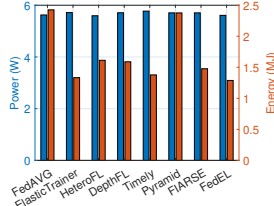

Figure 8: Memory overhead.

Figure 9: Power/energy consumption.

ing only a portion of the DNN model while freezing unselected layers and tensors, which minimizes memory allocation required for gradient backpropagation. In Figure 9, we observe little variation in power consumption across methods, as both Orin and Xavier operate at full power when their GPUs are active. However, for the same set of computational tasks, FedEL significantly reduces energy consumption. FedEL achieves an average reduction of 49.59% in total energy usage compared to FedAvg, primarily because it completes training in nearly half the time required by FedAvg.

**FedEL can adaptively select important tensors.** FedEL's performance is driven by its dynamic sliding-window mechanism and elastic tensor selection at runtime. We analyze these adaptive behaviors using a large-scale 100-device scenario with the Tiny ImageNet dataset. Figure 10 showcases representative devices from each of the four device types. As observed, the number of windows required to train the full model varies across devices due to their differing computational capabilities. Within each window, tensor selection is dynamically adjusted based on importance. For instance, if a tensor at the front is critical for model performance, FedEL can adaptively skip updating

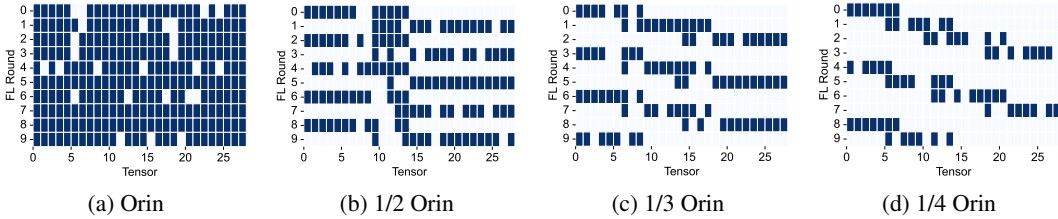

(a) Orin      (b) 1/2 Orin      (c) 1/3 Orin      (d) 1/4 Orin

Figure 10: Tensor selections during different FL rounds

a few less important tensors (with higher indices) to maintain the desired training speedup while preserving model effectiveness.

## 5.3 Ablation

We analyze how parameter settings influence FedEL using the small-scale practical 10-device scenario with the CIFAR10 dataset for image classification. Additional results can be found in the Appendix.

**Impact of balancing parameter $\beta$.** The balancing parameter $\beta$ in FedEL determines the weighting between local and global tensor importance during adjustment. Figure 11 shows how varying $\beta$ affects time-to-accuracy performance. A larger $\beta$ overemphasizes local tensor importance, reducing the influence of global model variations, while a smaller $\beta$ focuses solely on global variations, neglecting local data heterogeneity. As shown in Figure 11, when $\beta = 1$ (fully local focus) or $\beta = 0$ (fully global focus), FedEL's accuracy falls below that of FedAvg. In contrast, moderate values of $\beta$ (e.g., $\beta = 0.6$ or $\beta = 0.4$) outperform FedAvg by balancing local data heterogeneity with global model variations. This balance allows FedEL to effectively capture both local and global tensor importance, enhancing accuracy.

**Impact of runtime threshold $T_{th}$.** To ensure a fair comparison with other baselines, we set the training time threshold $T_{th}$ equal to the full model training time of the Orin (i.e., $T_{Orin}$). We vary $T_{th}$ to examine its impact on FedEL's performance, with the experiment stopping once the training time reaches the predefined value. As shown in Figure 12, a smaller $T_{th}$ slows down convergence. This is because slow clients must train the entire model, leading to more sliding-window movements, while fast clients also perform additional window sliding, increasing the overall training time and reducing efficiency.

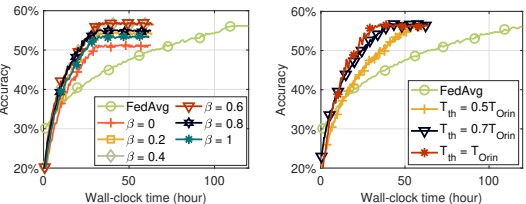

Figure 11: Impact of $\beta$. Figure 12: Impact of $T_{th}$.

**Sliding Window.** The sliding window consists of two processes: the front edge movement and the end edge movement, which define the window size and the range of selected important tensors. In each FL round, the front edge includes blocks with accumulated training time just above the runtime threshold $T_{th}$. As shown in Figure 12, reducing $T_{th}$ slows convergence, as more rounds are required to train the full model. The end edge movement reduces the window size by excluding unselected blocks.

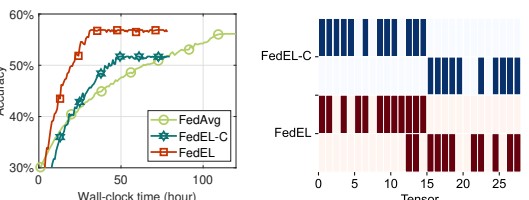

Figure 13: Time-to-accuracy of FedAvg, FedEL-C and FedEL.

Figure 14: Tensor selection illustration in FedEL-C and FedEL.

To assess its effectiveness, we compare it with a scenario where the end edge is directly moved to the current front edge (FedEL-C). As shown in Figure 13, FedEL-C results in lower accuracy than FedEL. The tensor selection examples in Figure 14 explain this: FedEL-C treats each window independently and does not adjust training tensors between consecutive windows, leading to accuracy degradation.

# 6 Conclusion

We introduced FedEL, a progressive training approach to address client heterogeneity in FL. To overcome the limitations of directly selecting important tensors, we propose two innovations: sliding-window training and local tensor importance adjustment. Sliding-window training enables FedEL to train the full DNN model by adjusting the front and end edges of the training window. Local tensor importance adjustment selects important tensors based on both local client data and global data importance. The results show that FedEL reduces wall clock training time (speeding up by $1.87\times$ to $3.87\times$) while achieving comparable or better accuracy and perplexity across various FL applications and DNN models.

# 7 Acknowledgments

The work of Letian Zhang and Bo Chen is partially supported by NSF under grant 2348279 and also supported by MTSU Stark Land project. The work of Jieming Bian, Lei Wang and Jie Xu is partially supported by NSF under grants 2433886, 2505381 and 2515982.

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

# A   The Algorithm of FedEL

In this paper, we introduce the sliding window training to address the first limitation and tensor importance adjustment to overcome the second limitation. We present a comprehensive window-based important tensor selection scheme implemented by FedEL, as outlined in Algorithm 1. Specifically, prior to the FL process, each client performs offline tensor time profiling for the DNN model (Lines 3-5), which is done only once. In each online FL round, once the client receives the broadcasted global model, it evaluates the tensor importance for the current global model (Line 8), calculates the global tensor importance (Line 9), and adjusts the local tensor importance accordingly (Line 10). Based on the previous round's training status, FedEL then slides or resets the window to ensure the entire DNN model is trained (Line 11). Once the window is fixed, ElasticTrainer is applied within the window to select important tensors, freeze unselected ones, and train only the selected tensors (Lines 12-13). Finally, the server aggregates the models from all clients and broadcasts the updated global model for the next FL round.

---
**Algorithm 1** FedEL
---
1: **Input:** Client set $\mathcal{N}$, training time threshold $T_{th}$, balance parameter $\beta$, DNN model $\boldsymbol{w}$.
2: **Output:** Trained model.
   $\triangleright$ Offline and only once
3: **for** Each client n: **do**
4:     TensorTimeProfiling($\boldsymbol{w}$).
5: **end for**
   $\triangleright$ Online
6: **for** Each FL round r: **do**
7:     **for** Each client n: **do**
8:         $\boldsymbol{I}_{n,r}$ = TensorImportanceEvaluation($\boldsymbol{w}_r$)
           $\triangleright$ Tensor Importance adjustment
9:         $\boldsymbol{I}^g$ = GetGlobalTensorImportance($\boldsymbol{w}_r, \boldsymbol{w}_{r-1}, \eta_n$)
10:        $\boldsymbol{I}_{n,r}$ = AdjustLocalTensorImportance($\boldsymbol{I}_{n,r}, \beta, \boldsymbol{I}^g$)
           $\triangleright$ Window Sliding
11:        $\Theta_{n,r}$ = SlideWindow($\boldsymbol{w}_r, T_{th}, \Theta_{n,r-1}$)
           $\triangleright$ Elastic Training
12:        $\boldsymbol{A}_{n,r}$ = SelectImportantTensor($\Theta_{n,r}, T_{th}, \boldsymbol{I}_{n,r}$)
13:        $\boldsymbol{w}_{n,r}$ = TrainImportantTensor($\boldsymbol{A}_{n,r}, \boldsymbol{w}_r$)
14:     **end for**
15:     $\boldsymbol{w}_{r+1}$ = Aggregate($\boldsymbol{w}_{n,r}$)                       $\triangleright$ Server side
16: **end for**

---

# B   Detailed Datasets and Baselines

**Baselines.** The following baselines are adopted for evaluation purposes:

(1) **FedAvg** [31] is the classic generic FL algorithm without accounting for system heterogeneity. Each client trains the same full DNN model.

(2) **ElasticTrainer** [14] is directly deployed into the local training clients of FedAvg framework.

(3) **HeteroFL** [7] facilitates training across heterogeneous devices by scaling the channels of convolutional layers to cater to diverse computation constraints.

(4) **DepthFL** [18] segments the model into sub-models of varying depths, distributing them to clients according to their computing capabilities.

(5) **PyramidFL** [25] aims to enhance time-to-accuracy by considering both data and system heterogeneity during binary client selection.

(6) **TimelyFL** [50] is a heterogeneity-aware asynchronous FL framework with adaptive partial training.

(7) **FIARSE** [32] dynamically masks the unimportant layers with adaptive partial training.

**Datasets, Models, and Tasks.** To demonstrate FedEL's effectiveness across tasks, datasets, and ML models, we evaluate FedEL on four real-world datasets designed for FL applications at different scales. To follow the realistic non-iid data in FL scenarios, we partition the datasets into different clusters using a Dirichlet distribution with $\alpha$ equals 0.1.

● **Image Classification.** The CIFAR10 dataset [19] consists of 60,000 colored images in 10 classes. The Tiny ImageNet dataset [23] contains 100000 images of 200 classes colored images. We evaluate the dataset with VGG16 [35] model.

● **Speech Recognition.** The Google Command speech dataset [40] covers 105,829 audio commands recordings. The data set is composed of 35 common words from the everyday vocabulary, such as "Yes", "No", "Up", and "Down". We evaluate the datasets with ResNet50 [11] model for a 35-class keyword spotting task.

● **Natural Language Processing.** Reddit [32] consists of comments from 1,660,820 users in the Reddit forum. In this dataset, we filter the users with less than 20 words in total and restrict to the 30k most frequently used words, as the same settings in the previous work [20]. Then, we fine turn the lightweight Albert [22] model for the next-word-prediction task. The performance is evaluated by the perplexity loss, which lower is better. It's worth noting that Reddit datasets inherently exhibits non-iid characteristics. We follow [43] to generate the blocks of the lightweight Albert model.

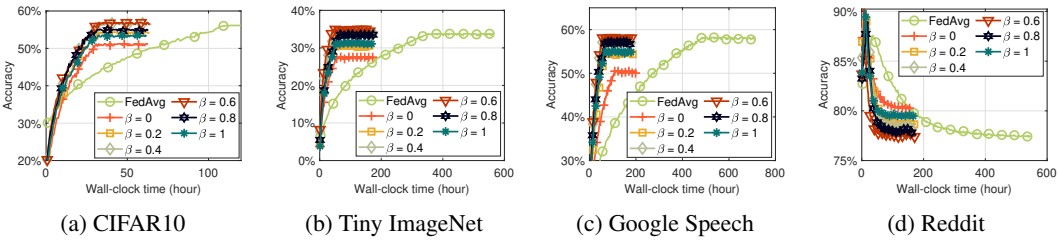

(a) CIFAR10      (b) Tiny ImageNet      (c) Google Speech      (d) Reddit

Figure 15: Impact of balancing parameter $\beta$ on four tasks.

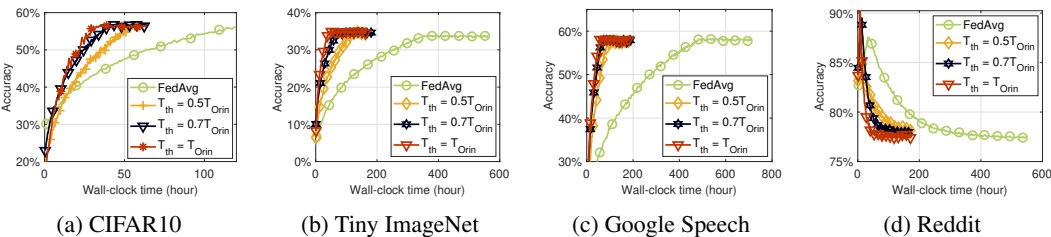

(a) CIFAR10      (b) Tiny ImageNet      (c) Google Speech      (d) Reddit

Figure 16: Impact of runtime threshold $T_{th}$ on four tasks.

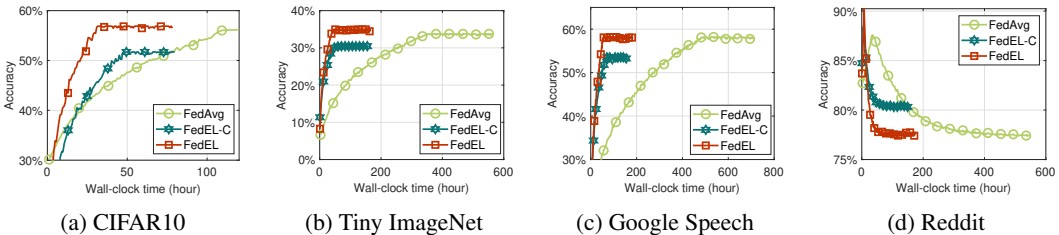

(a) CIFAR10      (b) Tiny ImageNet      (c) Google Speech      (d) Reddit

Figure 17: Time-to-accuracy of FedAvg, FedEL-C and FedEL on four tasks.

## B.1 Ablation

In the Ablation section of the main paper, we analyze the effect of parameter settings on FedEL using the CIFAR10 dataset. Here, we show the remaining ablation results for other three tasks in a large 100-device scenario.

**Impact of balancing parameter $\beta$.** Figure 15 illustrates the impact of varying $\beta$ on time-to-accuracy performance across the Tiny ImageNet, Google Speech, and Reddit datasets. In FedEL, the balancing

parameter $\beta$ controls the trade-off between local and global tensor importance during adjustment. A larger $\beta$ places greater emphasis on local tensor importance, reducing the influence of global model variations. Conversely, a smaller $\beta$ prioritizes global variations while neglecting local data heterogeneity. When $\beta = 1$ (fully local) or $\beta = 0$ (fully global), FedEL achieves lower accuracy than FedAvg. However, with moderate values ($\beta = 0.4$ or $\beta = 0.6$), FedEL outperforms FedAvg by effectively balancing local heterogeneity with global model updates. This balance enables FedEL to capture both local and global tensor importance, leading to improved accuracy.

**Impact of runtime threshold** $T_{th}$**.** Figure 16 illustrates how varying the runtime threshold $T_{th}$ affects performance across three additional tasks in a 100-device scenario. To ensure a fair comparison with baseline methods, we set $T_{th}$ equal to the full model training time on fastest device. We then vary $T_{th}$ to analyze its impact, stopping the experiment once the total training time reaches the predefined limit. As shown in Figure 16, a smaller $T_{th}$ slows convergence. This occurs because slow clients must train the entire model, requiring more sliding-window movements and fast clients also perform additional window sliding, increasing overall training time and reducing efficiency.

**Sliding Window.** The sliding window operates through two processes. Front edge movement: Expands the window by including blocks until their cumulative training time slightly exceeds $T_{th}$. End edge movement: Shrinks the window by excluding unselected blocks. As shown in Figure 17, reducing $T_{th}$ results in slower convergence, as more rounds are required to train the full model. To evaluate the effectiveness of end edge movement, we compare it with a variant called FedEL-C, where the end edge is immediately shifted to the current front edge. Figure 17 shows that FedEL-C leads to lower accuracy than FedEL, highlighting the importance of gradual end edge adjustments for maintaining model performance. This is because FedEL-C treats each window independently and does not adjust training tensors between consecutive windows, leading to accuracy degradation.

## B.2 Important Tensor Selection

In the main paper, we demonstrated tensor selection in a large-scale scenario with 100 devices, using the VGG16 model on the Tiny ImageNet dataset. Here, we present tensor selection results for additional tasks. Figure 18 illustrates tensor selection on VGG16 with the CIFAR10 dataset for representative Orin and Xavier devices. Figure 19 shows results for ResNet50 on the Google speech dataset, using representative devices from each of the four device types. Figure 20 presents tensor selection for fine-tuning the Albert model on the Reddit dataset. Specifically, we freeze the pre-trained albert-base-v2 model and train only the newly added output layers. As observed, the number of windows required to train the full model varies across devices due to their differing computational capabilities. Within each window, tensor selection is dynamically adjusted based on importance. For instance, if a tensor in an earlier layer is critical for model performance, FedEL can adaptively skip updating certain less important tensors (with higher indices). This ensures an optimal balance between training speedup and model effectiveness.

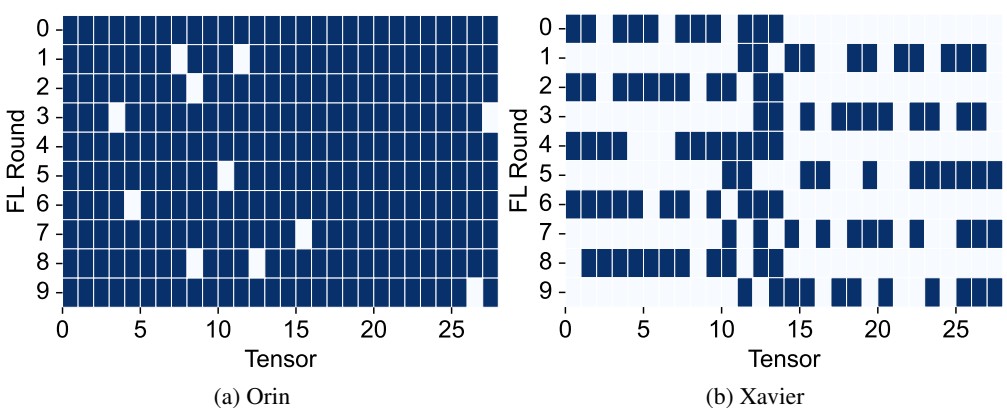

(a) Orin          (b) Xavier

Figure 18: Tensor selection of CIFAR10 dataset.

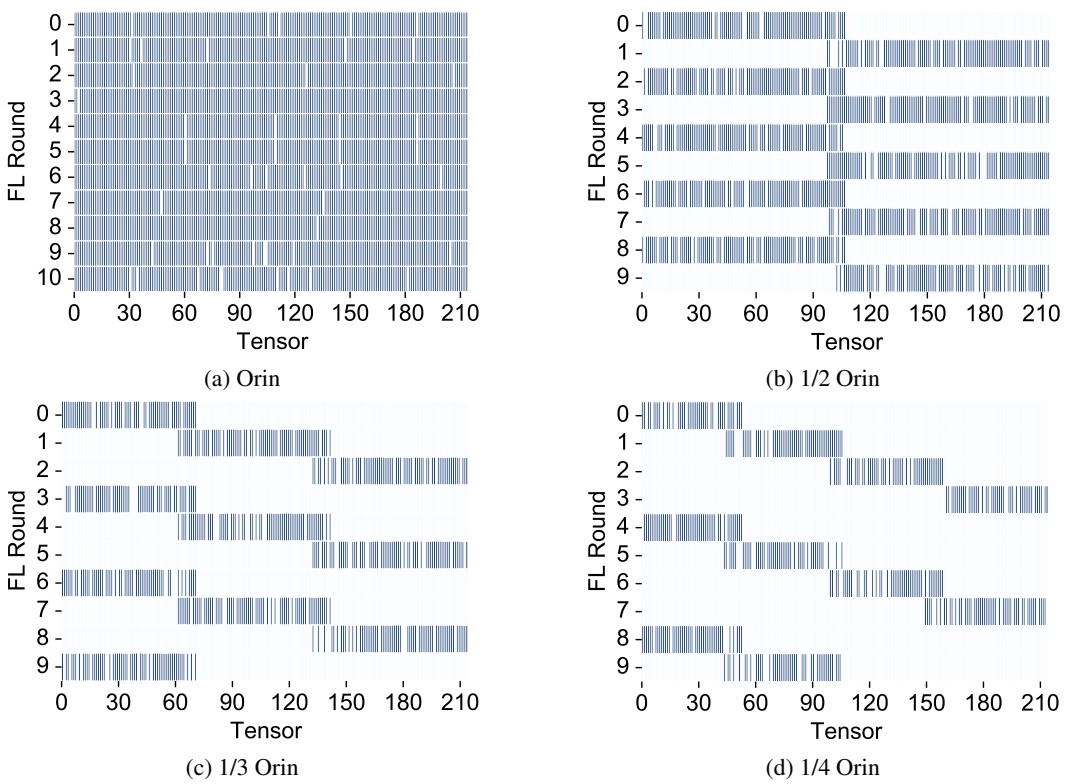

Figure 19: Tensor selection of Google Speech dataset.

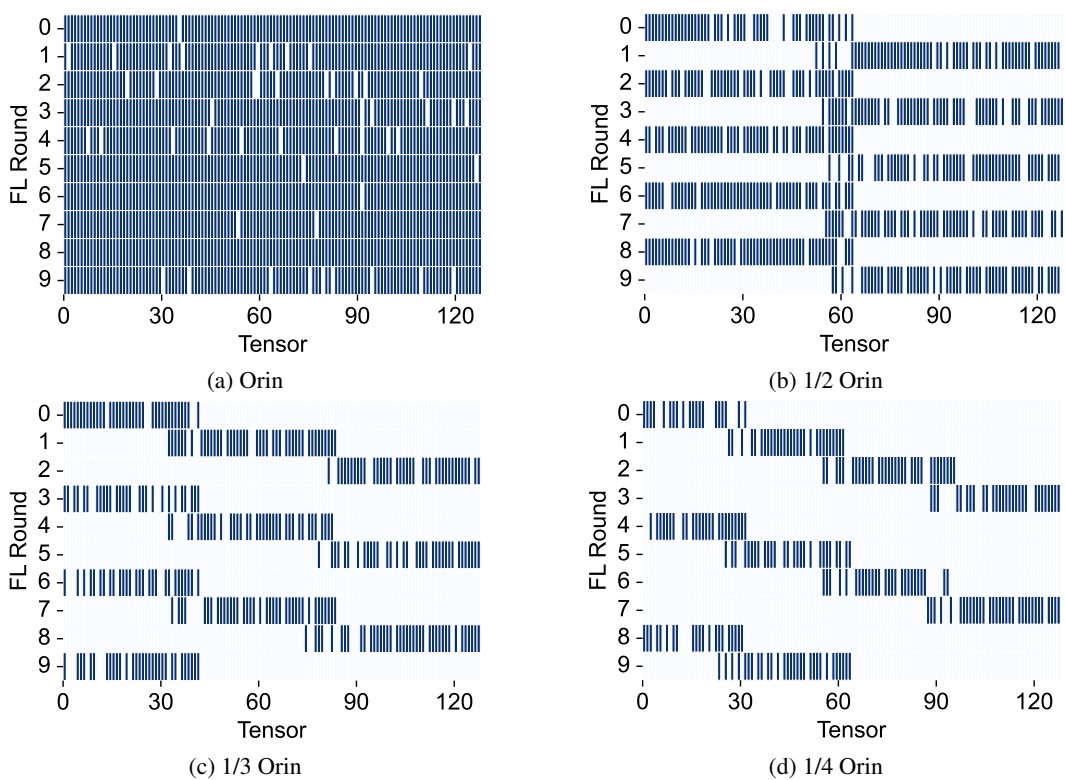

Figure 20: Tensor selection of Reddit dataset.

**B.3  How much does the training time deviate from the target time $T_{th}$?**

The differences in model architectures contribute to deviations between FedEL's training time and $T_{th}$. The table 2 presents the per-round average training time of FedEL compared to $T_{th}$. As observed, for convolutional networks (i.e., VGG16 and ResNet50), the deviation ranges from $3.2\%$ to $6.8\%$, whereas for the LLM model (i.e., Albert), the deviation is $18.9\%$. Despite these variations, FedEL significantly accelerates training compared to FedAvg full-model training, achieving a $1.87\times$ speedup in a small-scale practical edge device scenario and a $3.13\times$ to $3.87\times$ speedup in a large-scale simulation scenario.

Table 2: Deviation between the training time and $T_{th}$.

|  | CIFAR10 | Tiny ImageNet | Google speech | Reddit |
|---|---|---|---|---|
| FedEL | 38.2min | 45.1min | 54.9min | 48.6min |
| $T_{th}$ | 36.0min | 42.2min | 53.2min | 40.9min |
| Difference | 6.1% | 6.8% | 3.2% | 18.9% |
| FedAvg | 71.8min | 161.9min | 212.9min | 152.1min |
| Speedup | **1.87×** | **3.59×** | **3.87×** | **3.13×** |

**B.4  FedEL with particular algorithms which try to address any data non-IIDness.**

To assess FedEL's compatibility with aggregation algorithms beyond FedAvg, we integrated it with FedProx [27] and FedNova [38], both designed for non-IID data scenarios. Following their official implementations, we modified local updates and global aggregation to incorporate FedEL's adaptive tensor selection.

The table below compares the performance of FedProx/FedNova with and without FedEL on CIFAR10 dataset. As shown, FedEL is not restricted to FedAvg; it can be seamlessly integrated into other FL aggregation methods, leveraging their advantages while mitigating their limitations, particularly in heterogeneous device environments.

Table 3: Time-to-accuracy for combining FedProx and FedNova with our FedEL.

| Method | Acc | Time | Speedup |
|---|---|---|---|
| FedProx | 56.1% | 82.3h | N/A |
| FedProx + FedEL | 56.6% | 45.4h | 1.81× |
| FedNova | 66.3% | 84.7h | N/A |
| FedNova + FedEL | 66.1% | 47.8h | 1.77× |

**B.5  Statistical Comparison.**

To confirm the significance of our accuracy improvements, we provide a detailed statistical analysis, including confidence intervals. As shown in our box plot Figure 21, the confidence intervals indicate that our method maintains or exceeds accuracy with statistically significant improvements over baseline methods.

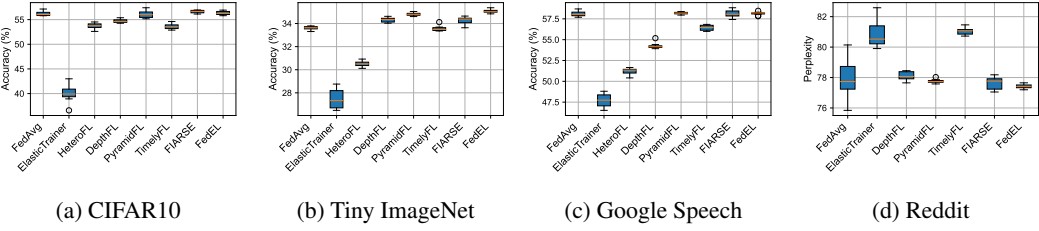

(a) CIFAR10          (b) Tiny ImageNet          (c) Google Speech          (d) Reddit

Figure 21: Accuracy statistical comparison.

Table 4: The value of the O1 term in the theoretical convergence upper bound for FedEL is analyzed for both cases: with and without rollback.

| Method | $O_1$ mean | $O_1$ std |
|---|---|---|
| Rollback | 63.06 | 8.62 |
| Not Rollback | 78.18 | 2.62 |

Table 5: Partition rate = 25%

| Method | Image Classif. Acc. / Time / Speedup | Speech Recog. Acc. / Time / Speedup | NLP Acc. / Time / Speedup |
|---|---|---|---|
| FedAvg | 33.76% / 563.1h / N/A | 58.04% / 709.8h / N/A | 77.48 / 546.4h /N/A |
| ElasticTrainer | 27.65% / 158.6h / 3.55× | 47.96% / 184.3h / 3.84× | 81.02 / 176.2h / 3.10× |
| HeteroFL | 30.56% / 248.2h / 2.26× | 51.47% / 265.9h / 2.66× | 80.11 / 206.1h / 2.65× |
| DepthFL | 34.14% / 198.3h / 2.83× | 54.23% / 207.4h / 3.42× | 78.08 / 212.4h / 2.57× |
| PyramidFL | 34.70% / 497.4h / 1.13× | 58.12% / 587.4h / 1.21× | 77.68 / 418.2h / 1.31× |
| TimelyFL | 33.53% / 198.1h / 2.84× | 56.49% / 193.2h / 3.67× | 80.91 / 177.6h / 3.07× |
| FIARSE | 33.98% / 191.5h / 2.94× | 58.13% / 198.2h / 3.58× | 77.31 / 191.0h / 2.86× |
| FedEL | 34.96% / 156.8h / 3.59× | 58.26% / 183.3h / 3.87× | 77.23 / 174.5h / 3.13× |

## B.6  Does the method rolling back blocks if necessary?

The rollback mechanism in sliding window training ensures that earlier layers can be retrained, allowing the model to refine learned representations rather than reinforcing suboptimal updates. This is particularly beneficial because deeper layers rely on feature representations from earlier layers. If earlier layers contain suboptimal representations, they can propagate errors throughout the network. By rolling back, the model can correct these errors and improve generalization, leading to more stable and effective learning.

In the convergence theorem of FedEL (Appendix E), tensor selection introduces an additional bias term O1. To analyze the impact of rollback, we designed two training scenarios:

1. Sliding window training with rollback, where layers can be revisited and updated.

2. Sliding window training without rollback, where the window shifts forward after a fixed number of rounds without revisiting earlier layers.

Table 4 presents the statistical values of the bias term O1 for both cases. As shown, the average value of O1 is smaller when rollback is allowed, compared to when it is not. This provides theoretical evidence that rolling back layers reduces the upper bound of convergence, leading to more stable and efficient training.

## C  The Improvements and Clarifications from NeurIPS Rebuttal

### C.1  Client Partitioning Scripts

- CIFAR-10 (10-client hardware deployment): We use full participation, where all 10 NVIDIA devices join every training round. This setting reflects a small-scale real-world deployment with stable device availability.

- Tiny-ImageNet, Google Speech Commands, and Reddit (100-client simulation): We adopt partial participation, where 25 clients are randomly selected out of 100 in each round (i.e., 25% participation rate). This follows common practice in large-scale FL simulations and models realistic device availability constraints.

To evaluate the impact of lower participation, we conducted experiments with a 10% participation rate. As shown in Table 5 and 6, all methods experienced slower convergence, leading to longer training time while maintaining similar accuracy. However, FedEL consistently achieves the highest accuracy and best efficiency across all tasks, confirming its advantage even under sparse client participation.

Table 6: Partition rate = 10%

| Method | Image Classif. Acc. / Time / Speedup | Speech Recog. Acc. / Time / Speedup | NLP Acc. / Time / Speedup |
|---|---|---|---|
| FedAvg | 33.75% / 782.4h / N/A | 58.01% / 987.3h / N/A | 77.45 / 764.2h / N/A |
| ElasticTrainer | 27.62% / 220.4h / 3.55× | 47.94% / 255.3h / 3.87× | 81.01 / 232.0h / 3.29× |
| HeteroFL | 30.52% / 344.8h / 2.27× | 51.45% / 392.2h / 2.52× | 80.09 / 288.7h / 2.65× |
| DepthFL | 34.12% / 275.7h / 2.84× | 54.21% / 310.5h / 3.18× | 78.07 / 298.9h / 2.56× |
| PyramidFL | 34.68% / 711.3h / 1.10× | 58.10% / 805.1h / 1.23× | 77.66 / 628.3h / 1.22× |
| TimelyFL | 33.50% / 278.4h / 2.81× | 56.45% / 267.2h / 3.69× | 80.89 / 233.3h / 3.28× |
| FIARSE | 33.96% / 269.2h / 2.91× | 58.11% / 271.4h / 3.64× | 77.28 / 251.6h / 3.04× |
| FedEL | 34.94% / 220.1h / 3.56× | 58.24% / 257.0h / 3.96× | 77.21% / 226.0h / 3.38× |

Table 7: Performance under different Jetson power modes.

| Method | Acc. | Time |
|---|---|---|
| FedAvg | 58.21% | 550.2h |
| FedEL | 59.35% | 174.3h |

## C.2  Jetson Power Modes

In our experiments, Jetson devices ran in MAXN mode to ensure consistency. We reran experiments under varied power settings (10W, 15W, MAXN). We measured training times under each mode and used this variability in the FedEL scheduling process. As shown in the Table 7, FedEL still outperforms FedAvg in both accuracy and training time. These results confirm that FedEL remains effective even in more diverse and realistic hardware environments.

## C.3  Communication and System Overhead

To clarify, our method does not add extra communication cost—in fact, it reduces it compared to FedAvg. This is because FedEL only uploads selected important tensors rather than the full model. As shown in the table, FedEL results in: 1. Lower communication time per round than FedAvg. 2. Communication taking up only a small part of the total training time. We also measured the runtime of FedEL's system modules (e.g., sliding window, tensor importance update, and selection). As shown in Table 8, the added overhead is minimal and has negligible effect on overall training time.

## C.4  Discussing potential extensions of our work to handle communication heterogeneity and integration with differential privacy.

Our study focuses on computational heterogeneity, a dominant bottleneck in mobile edge environments. In practice, training time on mobile devices far exceeds communication time, especially with modern 5G/WiFi networks. For example, ResNet50 ( 97.7 MB) takes 0.28–1.3 minutes to transmit over 10-45 Mbps uplink (e.g., AT&T, 2024), while training on an NVIDIA Jetson Xavier takes  38.3 minutes, making communication relatively negligible in our setting. Nonetheless, our method can incorporate communication heterogeneity. During offline profiling, we estimate tensor sizes. If client bandwidth is known, the tensor selection can be extended to:

$$\max_{\boldsymbol{A}} \boldsymbol{A} \cdot \boldsymbol{I}, \;\; s.t. \;\; T_{fw} + T_{bw}(\boldsymbol{A}) + T_{tx}(\boldsymbol{A}) \leq T_{th}. \tag{2}$$

where $T_{tx}(\boldsymbol{A})$ estimates transmission time. This enables FedEL to jointly optimize computation and communication under a unified latency constraint. Our method is fully compatible with privacy-preserving techniques like Differential Privacy (DP) and Secure Aggregation (SA), as FedEL operates at the system level and does not modify or interfere with model encryption or privatization.

## C.5  Detailing how our approach can be adapted for transformer models.

As detailed in Section B of our Supplementary Material, our implementation for Albert follows the block design from [43], placing early exits after each encoder-classifier stack. This supports

Table 8: System overhead.

| Method | Communication | Tensor processing | Average round time |
|--------|---------------|-------------------|--------------------|
| FedAvg | 2.45 min (3.2%) | N/A | 75.43 min |
| FedEL | 1.09 min (2.7%) | 0.97 min (2.4%) | 40.34 min |

variable-depth execution and is fully compatible with our sliding window mechanism. Regarding shared-weight layers, such as embeddings and grouped attention heads, we handle them as follows:
(1) Shared Embeddings: We profile the shared tensor as a single unit. Backward time is traced from the output back to the shared embedding, reflecting its impact on both input and output. The weight update time is counted only once since the tensor is updated once per iteration.
(2) Grouped Attention Heads: For attention layers with multiple trainable projections (query, key, value, output), we treat each as a separate tensor and profile their backward time individually. If some projections are shared, we track their computation cost jointly during backpropagation.

### C.6 More Related Works

Recurrent Early Exits (ReeFL) [24]: ReeFL enables early exits in LLMs during FL. Our method FedEL can work with ReeFL by applying tensor selection inside each early-exit block. We combined both by inserting exits into transformer layers and applying FedEL within each. As shown below, FedEL+ReeFL reduces training time while maintaining accuracy, confirming that the two approaches are complementary.
ScaleFL [15]: ScaleFL adapts model width/depth based on device profiles, but uses static selection via a meta-scheduler. In contrast, FedEL dynamically selects tensors in each round based on importance, allowing finer control and better adaptability.
NeurIPS 2023 [51] offers convergence guarantees but applies global pruning without local data or neuron importance awareness, limiting its adaptability at the client level.
TNNLS 2022 [17] supports distributed pruning but incurs significant communication overhead due to the need to transmit importance scores and pruning decisions each round.
DAC 2021 (Helios) [47] enables dynamic model adaptation but requires frequent communication and additional optimization overhead, especially for straggler mitigation.
TMC 2024 [13] does not perform partial training; instead, it trains the full model and uploads only a subset of parameters, which is orthogonal to our focus on efficient local training.
In contrast, FedEL performs lightweight, local tensor selection using an importance-based mechanism, incurs no extra communication, supports fine-grained dynamic adaptation, and avoids heavy optimization—offering a practical and scalable solution to device heterogeneity.

## D Limitation

To evaluate the effectiveness of FedEL, we conduct experiments in two settings: a small-scale practical setup using real edge devices, and a large-scale simulation. Due to hardware limitations, the practical setup includes only two types of edge devices. The large-scale simulation is then designed based on system measurements collected from these two devices. While this approach demonstrates promising results, it may face challenges when scaled to real-world environments with more extreme heterogeneity in client computational resources. Additionally, this work does not account for variations in client network bandwidth, which we plan to explore in future work.

## E Convergence Theorem.

We consider one server and N edge devices. Each device $n \in \mathcal{N} = \{1, 2, \ldots N\}$ has its own set of local data samples $\mathcal{D}_n$. In a supervised learning setting each device aims to find a learning model $\theta_n \in \mathbb{R}^d$, where $d_\theta$ denote the dimensions of the model. A mask $A_n \in \{0, 1\}^{d_\theta}$ is selected for each device $n \in \mathcal{N}$ based on the ElasticTrainer. During local update, each device n only updates those parameters in the global model that correspond to non-zero values of the masking vector $A_m$.

Let $w_n^i$ denote the local model of device $n$ at the beginning of local update iteration $i$ in training round $t$. The local model of device $n$ is updated using SGD as follows:

$$\boldsymbol{w}_n^{i+1}(t) = \boldsymbol{w}_n^i(t) - \eta(t)\boldsymbol{A}_n(t) \odot \nabla f_n(\boldsymbol{w}_n^i(t), b_n^i(t)), i = 1, \ldots, \tau \tag{3}$$

where $\odot$ denotes the element-wise product, $\eta(t)$ is the learning rate, $f_n(\cdot)$ is the loss function and $b_n^i(t)$ is the local batch sample chosen uniformly at random from the local dataset. After performing $\tau$ local update iterations, each device n sends its final model to the server.

$$\boldsymbol{w}_n(t) = \boldsymbol{w}_g(t) - \eta(t)\boldsymbol{A}_n(t) \odot \sum_{i=1}^{\tau} \nabla f_n(\boldsymbol{w}_n^i(t), b_n^i(t)) \tag{4}$$

In the aggregation step, we consider that the server aggregates the received final local models by taking the masking vectors of the devices into account. The global model for the next communication round can thus be determined through stable aggregation of unfrozen parameters, as follows:

$$\boldsymbol{w}_g(t+1) = \sum_{n \in \mathcal{N}} \boldsymbol{c}_n(t) \odot \boldsymbol{w}_n(t) \tag{5}$$

where $(\boldsymbol{c}_n(t))_k = \frac{(A_n(t))_k}{\sum_{n \in \mathcal{N}}(\boldsymbol{A}_n(t))_k}$ denotes the k-th tensor selection of mask $\boldsymbol{A}_n(t)$ at training round $t$. Using $(\boldsymbol{c}_n(t))_k$ in (5) indicates that the server only aggregates the updated parameters from the participating devices.

The analysis relies on the following assumptions, which are commonly used for obtaining the convergence rate of different FL algorithms in the literature.

**Assumption E.1.** The function $f_n(\boldsymbol{w}), n \in \mathcal{N}$ is L-smooth and satisfies:

$$||\nabla f_n(\boldsymbol{w}_n^i(t))||^2 \le 2L(f_n(\boldsymbol{w}_n^i(t) - f_n^*), n \in \mathcal{N}, i = 1, \ldots, \tau, \forall t \tag{6}$$

where $f_n^*$ denotes the minimum value of $f_n(\boldsymbol{w})$.

**Assumption E.2.** $\nabla f_n(\boldsymbol{w}_n^i(t), b_n^i(t))$ is an unbiased stochastic gradient of function $f_n(\boldsymbol{w})$. The variance of the masked stochastic gradients is bounded for each device $n \in \mathcal{N}$. We have

$$\mathbb{E}||\boldsymbol{c}_n(t) \odot \nabla f_n(\boldsymbol{w}_n^i(t), b_n^i(t)) - \boldsymbol{c}_n(t) \odot \nabla f_n(\boldsymbol{w}_n^i(t))||^2 \le \xi_n^2, n \in \mathcal{N}, i = 1, \ldots, \tau, \forall t \tag{7}$$

**Assumption E.3.** The expected squared L2-norm of the masked stochastic gradients for all the devices is uniformly bounded. We have

$$\mathbb{E}||\boldsymbol{A}_n(t) \odot \nabla f_n(\boldsymbol{w}_n^i(t), b_n^i(t))||^2 \le G^2, n \in \mathcal{N}, i = 1, \ldots, \tau, \forall t \tag{8}$$

**Lemma E.4.** *The following inequality holds for any vectors $x$ and $z \in \mathbb{R}^d$, for which there exists $Q > 0$ satisfying $|min_k(x \odot z)_k| \le Q$, and for any vector $y \in \mathbb{R}^d$.*

$$\langle x, y \odot z \rangle \le \max_k(y)_k \langle x, z \rangle + Q \left( d \max_k(y)_k - \sum_{k=1}^{d}(y)_k \right) \tag{9}$$

*where $\langle \cdot, \cdot \rangle$ denotes the inner product operator in $\mathbb{R}^d$.*

*Proof.* Given vectors $x$, $y$, and $z$, we form diagonal matrices X, Y and Z, respectively. Note that wee can write $\langle z, y \odot z \rangle$ as the form of the trace of matrices X, Y and Z product, i.e., $\langle z, y \odot z \rangle = Tr(XYZ)$. By using Theorem 3 in [8], we have the following inequality:

$$Tr(XYZ) \le \lambda_1(Y)Tr(XZ) - \lambda_d(XZ)(d\lambda_1(Y) - Tr(Y)) \tag{10}$$

where $\lambda_1(Y)$ and $\lambda_d(XZ)$ are the largest eigenvalue of matrix $Y$ and the smallest eigenvalue of matrix $XZ$, respectively. Since the considered matrices are diagonal, we have $\lambda_1(Y) = \max_k(y)_k$ and $\lambda_d(XZ) = \min_k(x \odot z)_k$. Hence, we have

$$\langle x, y \odot z \rangle \le \max_k(y)_k \langle x, z \rangle + \min_k(x \odot z)_k \left( d \max_k(y)_k - \sum_{k=1}^{d}(y)_k \right) \tag{11}$$

$$\le \max_k(y)_k \langle x, z \rangle + Q \left( d \max_k(y)_k - \sum_{k=1}^{d}(y)_k \right) \tag{12}$$

Since $d \max_k(y)_k - \sum_{k=1}^{d}(y)_k \ge 0$, by considering $|min_k(x \odot z)_k| \le Q$, Lemma E.4 is proved using the second inequality. $\square$

We define the term $\gamma_n(t) = \max_k(\boldsymbol{c}_n(t))_k$ to quantify the degree of device heterogeneity in the network. Note that in the full device participation scenario, $\frac{1}{N} \leq \gamma_n(t) \leq 1, n \in \mathcal{N}$. Then, the following Theorem E.5 shows that the convergence bound of employing the masks to address the device heterogeneity issue in FL. Then, the following Theorem E.5 shows that employing the masks to address the device heterogeneity issue in FL leads to a bias term in the convergence bound. However, it does not affect the convergence rate, which is similar to the observation in paper [41].

**Theorem E.5.** *Under Assumptions 1-3, and for smooth and non-convex loss functions, if the total number of communication rounds T is pre-defined and the learning rate $\eta(t)$ is smart enough such that $\eta(t) = \eta \leq \frac{1}{LN^2\tau}$, we have*

$$\frac{1}{T}\sum_{t=1}^{T}\mathbb{E}\|\nabla F(\boldsymbol{w}_g(t))\|^2 \leq \frac{2}{\eta\tau T}(F(w_g(1)) - F^*) + LN\tau\eta\sum_{n=1}^{N}\xi_n^2$$

$$+ 2\Psi\underbrace{\sum_{n=1}^{N}\left(d_\theta\gamma_n(t) - \sum_{k=1}^{d_\theta}(\boldsymbol{c}_n(t))_k\right)}_{O_1} + L^2\eta^2 G^2\frac{(\tau-1)(2\tau-1)}{6} \tag{13}$$

*where $\Psi$ is a constant satisfying $\max_k(\nabla f_n(\boldsymbol{w}_n^i(t),) \odot \nabla F(\boldsymbol{w}_g(t)))_k \leq \Psi, \forall n, i, t$. $F^* = F(\boldsymbol{w}^*)$, where $\boldsymbol{w}^*$ is the global optimal weight. $L$, $\xi_n^2$ and $G$ are constants defined in Assumptions 1-3.*

*Proof.* Considering the smoothness of $f_n(\boldsymbol{w}), n \in \mathcal{N}$, in each training round $t \geq 1$, we have

$$\mathbb{E}F(\boldsymbol{w}_g(t+1)) \leq \mathbb{E}F(\boldsymbol{w}_g(t)) + \mathbb{E}\langle\boldsymbol{w}_g(t+1) - \boldsymbol{w}_g(t), \nabla F(\boldsymbol{w}_g(t))\rangle + \frac{L}{2}\mathbb{E}\|\boldsymbol{w}_g(t+1)) - \boldsymbol{w}_g(t))\|^2 \tag{14}$$

We first find an upper bound for $\|\boldsymbol{w}_g(t+1)) - \boldsymbol{w}_g(t))\|^2$ as follows:

$$\mathbb{E}\|\boldsymbol{w}_g(t+1) - \boldsymbol{w}_g(t)\|^2 \overset{(a)}{=} \eta^2(t)\mathbb{E}\left\|\sum_{n=1}^{N}\boldsymbol{c}_n(t) \odot \sum_{i=1}^{\tau}\nabla f_n(\boldsymbol{w}_n^i(t), b_n^i(t))\right\|^2$$

$$\overset{(b)}{=} \eta^2(t)\,\mathbb{E}\underbrace{\left\|\sum_{n=1}^{N}\sum_{i=1}^{\tau}\boldsymbol{c}_n(t) \odot \nabla f_n(\boldsymbol{w}_n^i(t), b_n^i(t)) - \boldsymbol{c}_n(t) \odot f_n(\boldsymbol{w}_n^i(t))\right\|^2}_{M_1}$$

$$+ \eta^2(t)\underbrace{\left\|\sum_{n=1}^{N}\sum_{i=1}^{\tau}\boldsymbol{c}_n(t) \odot f_n(\boldsymbol{w}_n^i(t))\right\|^2}_{M_2} \tag{15}$$

where equality (a) results from (4) and (5). Equality (b) is obtained via basic equality $\mathbb{E}\|\boldsymbol{z}\|^2 = \mathbb{E}\|\boldsymbol{z} - \mathbb{E}\boldsymbol{z}\|^2 + \|\mathbb{E}\boldsymbol{z}\|^2$ for any random vector $\boldsymbol{z}$. By using Assumption E.2, we have obtain an upper bound of $M_1$ as follows:

$$M_1 = \mathbb{E}\left\|\sum_{n=1}^{N}\sum_{i=1}^{\tau}\boldsymbol{c}_n(t) \odot \nabla f_n(\boldsymbol{w}_n^i(t), b_n^i(t)) - \boldsymbol{c}_n(t) \odot f_n(\boldsymbol{w}_n^i(t))\right\|^2$$

$$\leq N\tau\sum_{n=1}^{N}\sum_{i=1}^{\tau}\mathbb{E}\left\|\boldsymbol{c}_n(t) \odot \nabla f_n(\boldsymbol{w}_n^i(t), b_n^i(t)) - \boldsymbol{c}_n(t) \odot f_n(\boldsymbol{w}_n^i(t))\right\|^2$$

$$\leq N\tau^2\sum_{n=1}^{N}\xi_n^2 \tag{16}$$

By considering the convexity of $\|\cdot\|^2$ and by using $\gamma_n(t) = \max_k (\boldsymbol{c}_n(t))_k$, we can obtain an upper bound of $M_2$ as follows:

$$M_2 = \left\| \sum_{n=1}^{N} \sum_{i=1}^{\tau} \boldsymbol{c}_n(t) \odot f_n(\boldsymbol{w}_n^i(t)) \right\|^2$$

$$\leq N\tau \sum_{n=1}^{N} \sum_{i=1}^{\tau} \left\| \boldsymbol{c}_n(t) \odot f_n(\boldsymbol{w}_n^i(t)) \right\|^2$$

$$\leq N\tau \sum_{n=1}^{N} \sum_{i=1}^{\tau} \gamma_n^2(t) \left\| f_n(\boldsymbol{w}_n^i(t)) \right\|^2 \tag{17}$$

By combining (15), (16) and (17), we have the following inequality:

$$\mathbb{E}\|\boldsymbol{w}_g(t+1) - \boldsymbol{w}_g(t)\|^2 \leq N\tau^2\eta^2(t) \sum_{n=1}^{N} \xi_n^2 + N\tau\eta^2(t) \sum_{n=1}^{N} \sum_{i=1}^{\tau} \gamma_n^2(t) \left\| f_n(\boldsymbol{w}_n^i(t)) \right\|^2 \tag{18}$$

Now, we aim to obtain an upper bound of $\mathbb{E}\langle \boldsymbol{w}_g(t+1) - \boldsymbol{w}_g(t), \nabla F(\boldsymbol{w}_g(t)) \rangle$. We have

$$\mathbb{E}\langle \boldsymbol{w}_g(t+1) - \boldsymbol{w}_g(t), \nabla F(\boldsymbol{w}_g(t)) \rangle$$

$$\overset{(a)}{=} \mathbb{E}\left\langle -\eta(t) \sum_{n=1}^{N} \sum_{i=1}^{\tau} \boldsymbol{c}_n(t) \odot \nabla f_n(\boldsymbol{w}_n^i(t), b_n^i(t)), \nabla F(\boldsymbol{w}_g(t)) \right\rangle$$

$$\overset{(b)}{=} \eta(t)\mathbb{E} \sum_{n=1}^{N} \sum_{i=1}^{\tau} \left\langle \boldsymbol{c}_n(t) \odot \nabla f_n(\boldsymbol{w}_n^i(t)), -\nabla F(\boldsymbol{w}_g(t)) \right\rangle$$

$$\overset{(c)}{\leq} \eta(t)\mathbb{E} \sum_{n=1}^{N} \sum_{i=1}^{\tau} -\gamma_n(t) \left\langle \nabla f_n(\boldsymbol{w}_n^i(t)), \nabla F(\boldsymbol{w}_g(t)) \right\rangle + \eta(t)\tau\Psi \sum_{n=1}^{N} \left( d_\theta \gamma_n(t) - \sum_{k=1}^{d_\theta} (\boldsymbol{c}_n(t))_k \right)$$

$$\overset{(d)}{\leq} -\eta(t) \sum_{i=1}^{\tau} \mathbb{E} \left\langle \frac{1}{N} \sum_{n=1}^{N} \nabla f_n(\boldsymbol{w}_n^i(t)), \nabla F(\boldsymbol{w}_g(t)) \right\rangle + \eta(t)\tau\Psi \sum_{n=1}^{N} \left( d_\theta \gamma_n(t) - \sum_{k=1}^{d_\theta} (\boldsymbol{c}_n(t))_k \right) \tag{19}$$

where equality (a) results from (4) and (5). Equality (b) follows from $\mathbb{E}\nabla f_n(\boldsymbol{w}_n^i(t), b_n^i(t)) = \nabla f_n(\boldsymbol{w}_n^i(t))$. Inequality (c) holds by using Lemma E.4. Inequality (d) follows from $\gamma_n(t) \geq \frac{1}{N}$.

To find an upper bound for $-\mathbb{E}\left\langle \frac{1}{N} \sum_{n=1}^{N} \nabla f_n(\boldsymbol{w}_n^i(t)), \nabla F(\boldsymbol{w}_g(t)) \right\rangle$, we first represent it as follows:

$$-\mathbb{E}\left\langle \frac{1}{N} \sum_{n=1}^{N} \nabla f_n(\boldsymbol{w}_n^i(t)), \nabla F(\boldsymbol{w}_g(t)) \right\rangle$$

$$= \frac{1}{2}\mathbb{E}\left\| \frac{1}{N} \sum_{n=1}^{N} (\nabla f_n(\boldsymbol{w}_n^i(t)) - \nabla f_n(\boldsymbol{w}_g(t))) \right\|^2 - \frac{1}{2}\mathbb{E}\left\| \frac{1}{N} \sum_{n=1}^{N} \nabla f_n(\boldsymbol{w}_n^i(t)) \right\|^2 - \frac{1}{2}\mathbb{E}\|\nabla f_n(\boldsymbol{w}_g(t))\|^2 \tag{20}$$

Then, $\mathbb{E}\left\| \frac{1}{N} \sum_{n=1}^{N} (\nabla f_n(\boldsymbol{w}_n^i(t)) - \nabla f_n(\boldsymbol{w}_g(t))) \right\|^2$ is bounded as follows:

$$\mathbb{E}\left\| \frac{1}{N} \sum_{n=1}^{N} (\nabla f_n(\boldsymbol{w}_n^i(t)) - \nabla f_n(\boldsymbol{w}_g(t))) \right\|^2$$

$$\overset{(a)}{\leq} \frac{1}{N} \sum_{n=1}^{N} \mathbb{E}\left\| \nabla f_n(\boldsymbol{w}_g(t)) - \nabla f_n(\boldsymbol{w}_n^i(t)) \right\|^2$$

$$\overset{(b)}{\leq} \frac{L^2}{N} \sum_{n=1}^{N} \mathbb{E}\left\| \boldsymbol{w}_g(t) - \boldsymbol{w}_n^i(t) \right\|^2 \tag{21}$$

where inequality (a) results from the convexity of $\|\cdot\|^2$. Inequality (b) results from Assumption E.1. Now, we aim to bound $\mathbb{E}\|\boldsymbol{w}_g(t) - \boldsymbol{w}_n^i(t)\|^2$ for $i = 2, \ldots \tau$. By using (3), we have

$$
\mathbb{E}\|\boldsymbol{w}_g(t) - \boldsymbol{w}_n^i(t)\|^2
$$

$$
= \mathbb{E}\left\|\eta(t)\boldsymbol{A}_n(t) \odot \sum_{j=1}^{i-1} \nabla f_n(\boldsymbol{w}_n^j(t), b_n^j(t))\right\|^2
$$

$$
\leq \eta^2(t)(i-1)\sum_{j=1}^{i-1} \mathbb{E}\left\|\boldsymbol{A}_n(t) \odot \nabla f_n(\boldsymbol{w}_n^j(t), b_n^j(t))\right\|^2
$$

$$
\leq \eta^2(t)(i-1)^2 G^2 \tag{22}
$$

where the last inequality results from Assumption E.3, By combining (21) and (22), we have

$$
\mathbb{E}\left\|\frac{1}{N}\sum_{n=1}^{N}(\nabla f_n(\boldsymbol{w}_n^i(t)) - \nabla f_n(\boldsymbol{w}_g(t)))\right\|^2 \leq L^2 \eta^2(t)(i-1)^2 G^2 \tag{23}
$$

By combining (14) and (14) to (23), we have

$$
\mathbb{E}F(\boldsymbol{w}_g(t+1)) \leq \mathbb{E}F(\boldsymbol{w}_g(t)) + \frac{L}{2}N\tau^2\eta^2(t)\sum_{n=1}^{N}\xi_n^2 + \eta(t)\tau\Psi\sum_{n=1}^{N}\left(d_\theta\gamma_n(t) - \sum_{k=1}^{d_\theta}(\boldsymbol{c}_n(t))_k\right)
$$

$$
- \frac{\eta(t)\tau}{2}\mathbb{E}\left\|\nabla F(\boldsymbol{w}_g(t))\right\|^2 + L^2\eta^3(t)G^2\frac{\tau(\tau-1)(2\tau-1)}{12}
$$

$$
- \frac{\eta(t)}{2}\sum_{n=1}^{N}\sum_{i=1}^{\tau}\left(\frac{1}{N} - LN\tau\gamma_n^2(t)\eta(t)\right)\|\nabla F(\boldsymbol{w}_g(t))\|^2 \tag{24}
$$

Since $\eta(t) = \eta \leq \frac{1}{LN^2\tau}$, we have last term $-\frac{\eta(t)}{2}\sum_{n=1}^{N}\sum_{i=1}^{\tau}\left(\frac{1}{N} - LN\tau\gamma_n^2(t)\eta(t)\right)\|\nabla F(\boldsymbol{w}_g(t))\|^2 \leq 0$. By rearranging the terms in (24), we obtain

$$
\mathbb{E}\left\|\nabla F(\boldsymbol{w}_g(t))\right\|^2 \leq \frac{2}{\eta\tau}(\mathbb{E}F(\boldsymbol{w}_g(t)) - \mathbb{E}F(\boldsymbol{w}_g(t+1))) + LN\tau\eta\sum_{n=1}^{N}\xi_n^2
$$

$$
+ 2\Psi\sum_{n=1}^{N}\left(d_\theta\gamma_n(t) - \sum_{k=1}^{d_\theta}(\boldsymbol{c}_n(t))_k\right) + L^2\eta^2 G^2\frac{\tau(\tau-1)(2\tau-1)}{12} \tag{25}
$$

Finally, we multiply both sides of (25) by $\frac{1}{T}$ and sum over $t = 1, \ldots, T$. Then, Theorem E.5 is concluded by considering that the first term on the right-hand side of (25) is a telescoping series. We have

$$
\frac{2}{\eta\tau T}\sum_{t=0}^{T}(\mathbb{E}F(\boldsymbol{w}_g(t)) - \mathbb{E}F(\boldsymbol{w}_g(t+1))) = \frac{2}{\eta\tau T}(F(\boldsymbol{w}_g(1)) - \mathbb{E}F(\boldsymbol{w}_g(T+1)))
$$

$$
\leq \frac{2}{\eta\tau T}(F(\boldsymbol{w}_g(1)) - F^*) \tag{26}
$$

where the last inequality is obtained by considering that $\mathbb{E}F(\boldsymbol{w}_g(t+1)) \geq F^*$. $\qquad\square$

