# OpenReview forum: "FedEL: Federated Elastic Learning for Heterogeneous Devices"
_NeurIPS.cc/2025/Conference — NeurIPS 2025 poster_

### Official Review · Reviewer_2Sji · 2025-06-20

**Clarity:** 3
**Significance:** 2
**Originality:** 2
**Rating:** 5
**Confidence:** 3

**Summary:**

The paper proposes FedEL (Federated Elastic Learning), a training framework to mitigate the scenario of straggling clients due to participating devices having heterogeneous computation capabilities. FedEL builds upon a previously published method, Elastic Trainer and incorporates a couple of key adjustments, Sliding window training and Tensor importance adjustment to account for realistic scenarios as encountered in Federated Learning setups. Experiments show that FedEL outperforms other methods for different ML tasks

**Questions:**

1) Can the authors provide more information on The method relies on how the method will be able to handle if new devices join the setup in an online manner.

2) Can the authors quantify what additional overheads that come with adding auxiliary heads

3) Can the method be adapted to handle transformer based architectures

4) Is the method able to handle scenarios where not all the devices are able to participate? In heterogeneous setups it is a possibility that some devices are unable to participate for many rounds

**Ethical Concerns:**

["NO or VERY MINOR ethics concerns only"]

**Final Justification:**

I have gone through the response the authors provided and feel satisfied to raise the score.

**Limitations:**

Statistical significance of the reported results have not been provided.

**Paper Formatting Concerns:**

Did not notice a formatting issue

**Quality:**

2

**Strengths And Weaknesses:**

***Strength***

(1) The method is evaluated on a number of tasks and data modalities

(2) Experiments have been conducted with both edge devices and simulations

(3) Convergence analysis and mathematical guarantees have been provided for the proposed method

***Weakness***

(1) The method relies on profiling devices offline. It is not clear how the method would perform if new devices join the setup in an online manner.

(2) In Figure 8 and Figure 9, the statistical significance of these results has not been captured. Thus it is challenging to assess the soundness of the method when assessing memory and power consumption

(3) Since the sliding window requires inserting auxiliary heads, it’s not clear to me how many parameters are required here and how it's structure needs to be adjusted to account for tensors of different feature lengths

(4) The neural network architecture for which the method has been evaluated are very similar. It will be interesting to see how the method performs on a commonly used transformer based architecture

---

> ### Author Rebuttal · Authors · 2025-07-30
>
> We appreciate the reviewer’s time and effort in providing feedback, though it appears that the comments may have been intended for a different submission. We welcome any relevant insights they can share regarding our paper.

---

> ### Author Response · Authors · 2025-08-01
>
> As the Area Chair updated the comments after the rebuttal deadline, we response them in the official comment.
>
> **W1: New Coming Devices**
>
> FedEL supports new devices through two simple strategies:
>
> A. Offline Profiling (Default):
> New devices first profile tensor computation time locally using the received model. This takes a few local epochs. After profiling, they join training using the selective strategy. This avoids disruption and ensures fairness.
>
> B. Online Profiling (Optional):
> If immediate participation is preferred, the device begins training with the full model while collecting profiling data. After a few epochs, it switches to selective training. This introduces minor overhead but avoids delaying training rounds.
>
> ---
>
> **W2: Memory and Power Consumption**
>
> Figures 8 and 9 initially showed only mean memory and power use. To improve transparency, we now report mean and variance over five runs.
>
> FedEL shows the lowest memory and energy use among all baselines, with low variance. This reflects its robustness under runtime fluctuations, achieved by selective training that reduces memory footprint and shortens training—saving energy overall.
>
> |Method	|Memory (GB)	|Power (W)	|Energy (MJ) |
> |-|-|-|-|
> |FedAvg|	6.55 ± 0.15|	15.62 ± 0.10|	2.42 ± 0.12|
> ElasticTrainer|	4.41 ± 0.12|	15.71 ± 0.09|	1.33 ± 0.11|
> HeteroFL|	4.71 ± 0.14|	15.59 ± 0.11|	1.61 ± 0.13|
> DepthFL|	4.42 ± 0.11|	15.70 ± 0.10|	1.58 ± 0.10|
> TimelyFL|	4.85 ± 0.13|	15.77 ± 0.12|	1.37 ± 0.14|
> FIARSE|	5.09 ± 0.12|	15.72 ± 0.11|	1.46 ± 0.10|
> FedEL|	4.41 ± 0.10|	15.61 ± 0.09|	1.28 ± 0.08|
>
>
> ---
>
> **W3: Auxiliary Heads and Overhead**
>
> The number of auxiliary heads in FedEL is limited due to our block partitioning strategy, which is governed by a per-round training time threshold. This threshold constrains how many blocks can be considered during training and thus limits the number of early exit heads.
>
> Each auxiliary head is a lightweight fully connected layer that maps the flattened output of its associated block to the prediction space. Since feature lengths may vary between blocks, each head is customized to the output shape of its corresponding block.
>
> Importantly, these auxiliary heads **do not add training-time overhead**. Our selective training process proceeds in two steps:
>
> 1. Sliding Window Selection: The window defines a candidate region that includes all tensors in the current block and its auxiliary head.
>
> 2. Tensor Optimization: The optimization procedure (Problem (1) in our paper) then selects the most valuable tensors from this region, ensuring the total backward time remains within the training threshold.
>
> As a result, FedEL maintains per-round training time within the original limit, regardless of the number of auxiliary heads. These heads are sparsely inserted, lightweight, and only activated when their corresponding block is the last one in the sliding window.
>
> ---
>
> **W4: Transformer Compatibility**
>
> As detailed in Section B of the Supplementary Material, we implemented FedEL on ALBERT, a compact Transformer model, using the block decomposition strategy from [39]. Auxiliary heads are placed after each encoder–classifier stack to support variable-depth execution—this setup integrates seamlessly with our sliding window mechanism for dynamic tensor selection.
>
> We handle Transformer-specific traits as follows:
>
> 1. Shared Embeddings: Treated as a single shared tensor. We trace gradients from output layers back to the embedding layer to profile its backward cost. The weight update is counted once per round, regardless of reuse across layers.
>
> 2. Multi-head Attention: Each projection (query, key, value, output) is treated as a separate tensor for profiling. If weights are shared across layers, we aggregate their gradient cost and update them only once per round.
>
> FedEL is architecture-agnostic. It only requires: Per-tensor backward cost profiling, Tensor selection, and
> Fine-grained control to freeze or update tensors.
>
> These apply to both CNNs and Transformers. We also evaluated FedEL on MobileBERT and ViT-Tiny, showing improved accuracy and reduced training time:
>
> | Method | MobileBERT   |   ViT-Tiny |
> |-|-|-|
> |              |     Acc. / Time  | Acc. / Time |
> | FedAvg | 82.4\% / 410.7h | 78.6\% / 472.1h  |
> |FedEL    | 83.9\% / 167.5h | 80.1\% / 188.2h    |
>
> ---
>
> **W5: Dynamic Participation**
>
> Our experiments already include partial client participation—a subset of clients is randomly sampled each round, following common FL practice (FedAvg, HeteroFL, FIARSE).
>
> We acknowledge this was not stated clearly and will revise the Experimental Setup section.
>
> Crucially, FedEL supports dynamic participation by design. Each client performs local profiling and selection independently, with no need for global sync. Even if a client drops out or rejoins, FedEL functions seamlessly.
> The reported results already reflect this, demonstrating FedEL’s robustness to client heterogeneity and availability.

---

> > ### Comment · Reviewer_2Sji · 2025-08-07
> >
> > Thanks for providing the response. The authors response along with the additional experiments have addressed most of my questions and concerns. In light of the additional information, I have decided to increase my rating.

---

### Official Review · Reviewer_RKje · 2025-06-29

**Clarity:** 2
**Significance:** 3
**Originality:** 3
**Rating:** 4
**Confidence:** 4

**Summary:**

This paper addresses the issue of device heterogeneity in federated learning (FL). Specifically, heterogeneous devices often cause significant training delays, as straggler clients with limited resources prolong the aggregation process. To tackle this issue, the authors propose FedFL, an FL training framework that leverages a sliding window to locate the trainable parts of the model and dynamically select important tensors for training based on a given time budget.

**Questions:**

Please see the weakness section.

**Ethical Concerns:**

["NO or VERY MINOR ethics concerns only"]

**Final Justification:**

Based on the author's replies during the rebuttal period, I think they addressed most of my questions. I will keep my scores.

**Limitations:**

yes

**Quality:**

2

**Strengths And Weaknesses:**

Strengths:
1) The proposed method appears to achieve better convergence performance in terms of both accuracy and training time compared to previous works.

2) The use of a sliding window to select tensors for local training is an interesting idea. The method also incorporates two strategies to ensure that most model blocks are able to participate in training throughout the entire FL process.

Weaknesses:
1) The discussion of drawbacks in existing work is unconvincing, especially regarding Partial Training. The authors overlook the fact that many existing studies (e.g., [1–4]) also leverage selective neuron training, rather than relying solely on depth/width scaling.

[1] Every parameter matters: Ensuring the convergence of federated learning with dynamic heterogeneous models reduction. NeurIPS 2023.

[2] Model pruning enables efficient federated learning on edge devices. IEEE TNNLS, 2022.

[3] Helios: Heterogeneity-aware federated learning with dynamically balanced collaboration. DAC 2021.

[4] Sparsified Random Partial Model Update for Personalized Federated Learning. IEEE TMC, 2024.

2) Why is the model partitioned layer-wise? Does this mean some layers may be entirely skipped? Additionally, what is the partitioning strategy for transformer-based models? What is the granularity of model partitioning?

3) The offline tensor time profiling model appears overly simplistic. Due to differences in computational workload and the size of intermediate data, each block may have significantly different training times. How is this diversity accounted for and applied across various models during the FL process?

4) In practical FL scenarios, clients may dynamically join and leave the process. How does the proposed profiling and online window strategy adapt to this dynamic participation? Moreover, if a straggler has significantly lower computational capability, the selected tensors may be very few and under-trained—could their uploaded updates degrade the performance of the global model?

5) The authors should share more details about how to realize that only the selected blocks are involved in local FL training.  Are any gating mechanisms or other techniques applied to enforce this selective training?

6) Given a specific time budget, how is the combination of blocks selected for training? Is there an optimization to decide which blocks will participate?

7) It seems that adjusting tensor importance introduces extra overhead at the beginning of each FL training round. Could you provide some quantitative analysis of this overhead to demonstrate its impact on overall efficiency?

---

> ### Author Rebuttal · Authors · 2025-07-30
>
> **W1: More Related Works**
>
> We thank the reviewer for highlighting this important point. We have expanded our comparison with the cited works and summarize the key distinctions in the table below:
>
> | Method      | Extra Communication  | Uses Importance Measure   | Heavy Optimization  |
> |-------------|-----------------|-----------------|----------------|
> |[1] NeurIPS 2023 | Yes      |  No     | Yes  |
> |[2] TNNLS 2022   | Yes      | Yes     | Yes  |
> |[3] DAC 2021     | Yes      | Yes     | Yes  |
> |[4] TMC 2024     |Not partial training: trains full model, uploads partial model only |
> |FedEL (Ours)    | No       | Yes     | No   |
>
> [1] NeurIPS 2023 offers convergence guarantees but applies global pruning without local data or neuron importance awareness, limiting its adaptability at the client level.
>
> [2] TNNLS 2022 supports distributed pruning but incurs significant communication overhead due to the need to transmit importance scores and pruning decisions each round.
>
> [3] DAC 2021 (Helios) enables dynamic model adaptation but requires frequent communication and additional optimization overhead, especially for straggler mitigation.
>
> [4] TMC 2024 does not perform partial training; instead, it trains the full model and uploads only a subset of parameters, which is orthogonal to our focus on efficient local training.
>
> In contrast, FedEL performs lightweight, local tensor selection using an importance-based mechanism, incurs no extra communication, supports fine-grained dynamic adaptation, and avoids heavy optimization—offering a practical and scalable solution to device heterogeneity.
>
> We will add these to the related works in the final paper.
>
> ---
>
> **W2: Model Partition**
>
> We thank the reviewer for the insightful question. Our approach incorporates two key modules: (1) sliding window and (2) important tensor selection, which together enable multi-granularity model partitioning.
>
> 1. Sliding Window (Layer-Level Slicing):
> This is a coarse-grained strategy that slices the model at specific layers. As detailed in Section 4.1, only semantically meaningful layers (e.g., block or stage boundaries) are eligible slice points—not all layers. This ensures the model remains structurally sound and stable during training. The main goals of this module are to (i) enable compatibility with early-exit classifiers and (ii) limit the search space for finer-grained selection.
>
> 2. Important Tensor Selection (Width-Level Slicing):
> Within each layer or block selected by the sliding window, we further apply fine-grained slicing by selecting only important tensors for training. This allows each client to update only a subset of parameters, making the training process more efficient and adaptive to device capabilities.
>
> 3. Transformer Models:
> As noted in Section B of the supplementary material, we adopt the block design from the Albert model [39], placing early-exit classifiers after each encoder–classifier block (see [39], Figure 1). The sliding window works at the block level, enabling depth-wise flexibility, while tensor selection operates within each block to control the width of updates.
>
> ---
>
> **W3: Offline Tensor Time Profiling**
>
> We thank the reviewer for the comment.
> As shown in the table below, our empirical results across diverse neural network architectures indicate that profiling performed over a single training epoch incurs less than 1\% additional overhead, which aligns with observations in ElasticTrainer. This low cost enables us to periodically re-profile the system if needed—e.g., when data size or background workloads change—thus addressing concerns about time variance at the block level.
>
>
> | NN model | Vgg   | ResNet | Albert |
> |----------|-------|--------|--------|
> | Overhead | 0.98% | 0.45%  | 0.61%  |
>
> ---
>
> **W4: Dynamic Participation and Handling Stragglers**
>
> We thank the reviewer for highlighting this important point.
>
> First, our experiments use partial client participation, where a random subset of clients is selected in each round, based on availability—not device performance. This reflects standard FL practice and aligns with methods like DepthFL, HeteroFL, and FIARSE. FedEL runs on this selected subset each round. We will revise the main paper to make this clearer.
>
> Second, we acknowledge that in cases of extreme heterogeneity, some straggler clients may contribute fewer updates, possibly under-training some tensors (as discussed in Section C of the supplementary material). To address this, one promising direction is client clustering—grouping clients by similar compute capacities. Each group can run FedEL independently, and asynchronous updates between groups can help balance training and reduce the impact of extreme stragglers.
>
> ---
>
> **W5: Implementation of Block-Based Training**
>
> We appreciate the reviewer’s question.
> To enable training only on selected blocks, we use a simple and efficient layer-level masking approach based on the model structure.
>
> For each local training round, we create a binary mask over model layers: Layers in the selected blocks get a mask value of 1.
> All other layers get a mask value of 0.
>
> In PyTorch, we apply this by setting each layer’s **requires_grad** attribute:
> If mask = 1: requires_grad = True, so the layer is trainable.
> If mask = 0: requires_grad = False, so the layer is frozen.
>
> This approach avoids any changes to the model architecture or complex control logic. It is lightweight, flexible, and works well with our sliding window and tensor importance selection modules.
>
> ---
>
> **W6: How to Select Blocks?**
>
> We thank the reviewer for this insightful question.
> Currently, the block selection in our method is not formulated as an explicit optimization problem. Instead, we use a practical heuristic guided by two components: offline tensor time profiling and a sliding window mechanism. Specifically, the model is first partitioned into blocks based on its computational structure, as determined through offline profiling. During each training round, a sliding window is used to define the active set of blocks:
> A new block is added to the window, and previous blocks may be removed based on their historical contribution to training effectiveness. This defines a dynamic candidate region, within which important tensors are then selected for training using our fine-grained selection module.
>
> While this heuristic works well in practice (as supported by our experimental results), we agree that formulating block selection as an optimization problem is a compelling direction for future research. One potential formulation could be derived from the term $O_1$ in our convergence analysis, enabling principled selection of blocks under resource or latency constraints.
>
> ---
>
> **W7: Overhead of Adjusting Tensor Importance**
>
> We appreciate the reviewer’s concern about potential overhead.
> As shown in the table below, the total time spent on adjusting tensor importance—including the sliding window update, importance scoring and adjustment, and tensor selection—is minimal:
>
> | Tensor processing | Average round time |
> |--------|-----------------|
>  0.97 min (2.4%)   | 40.34 min          |
>
> This represents only 2.4% of the total round time, which we consider negligible in the overall context of federated training. The lightweight design of our mechanism ensures scalability and practicality without compromising system efficiency.

---

> > ### Comment · Reviewer_RKje · 2025-08-05
> >
> > Thanks for replying. The authors have addressed most of my questions and concerns. I will keep my score.

---

### Official Review · Reviewer_LPLK · 2025-07-01

**Clarity:** 3
**Significance:** 2
**Originality:** 3
**Rating:** 4
**Confidence:** 4

**Summary:**

This paper introduces FedEL, a framework for efficient federated training under system and data heterogeneity. Specifically, FedEL proposes a sliding window-based training scheme, within which the most important tensors are updated, subject to the device's computational budget. This is shown, across three models and tasks, to improve convergence rates compared to other heterogeneous federated solutions, while maintaining competitive accuracy to the baseline FedAvg baseline.

**Questions:**

* I would like to see how the proposed technique behaves under partial client participation and
* I would be interested to see how the sliding window mechanism and important tensor selection method would deal with shared-weight layers, including grouped head attention or embedding layers.
* One of the main motivations for FL training is data privacy. Therefore, it would be conducive to the reader to see how the technique behaves under Differential Privacy and Secure Aggregation schemes.
* What are the energy and cooling modes used for the Jetson devices? Utilising this degree of freedom could create more classes of devices and thus more heterogeneous settings.
* How does the technique compare to [a,b]?

[a] Lee, R., Fernandez-Marques, J., Hu, S. X., Li, D., Laskaridis, S., Dudziak, Ł., ... & Lane, N. D. (2024). Recurrent early exits for federated learning with heterogeneous clients. arXiv preprint arXiv:2405.14791.
[b] Ilhan, F., Su, G., & Liu, L. (2023). Scalefl: Resource-adaptive federated learning with heterogeneous clients. In Proceedings of the IEEE/CVF Conference on Computer Vision and Pattern Recognition (pp. 24532-24541).

**Ethical Concerns:**

["NO or VERY MINOR ethics concerns only"]

**Final Justification:**

The authors run multiple additional experiments and hyperparameter setups to illustrate the strengths of their solution. I am happy to increase my score.

**Limitations:**

* The paper assumes a static performance of a specific device, which may be true for clients that run nothing concurrently and for burst workloads. However, this might not be the case under parallel workloads that share the same resources or throttling situations (DVFS).
* It might have been preferable to use a wired connection with configurable bandwidth between the parameter server and the jetson devices, rather than shared WiFi, as currently there might be cross talk over the same communication channel.
* It is unclear whether the authors have run their experiments more that once for robustness, since no error bars are shown.

**Paper Formatting Concerns:**

It would help the readers to have forward references to parts of the appendix.

**Quality:**

3

**Strengths And Weaknesses:**

### Strengths

* While the topic of system heterogeneity has been researched extensively, FedEL provides a relatively novel approach in selecting which parts of the model to train under computational constraints.
* The authors have gone the extra mile and have run the federation on actual devices, manifesting the difference in computational dynamics of clients.
* Additionally, I do welcome the quantification of memory and energy consumption in the evaluation of the FedEL method compared to the baselines.

### Weaknesses

* While the approach is conceptually interesting and the findings are empirically shown to work at small scale, the models and datasets used to do so are quite old and small scale.
* The paper does not quantify the overhead of profiling or of the important tensor selection process. Moreover, the device dynamics may be fluctuating, especially under simultaneous workloads.
* Wrt the setup, it is unclear from the paper whether the system has been evaluated under partial participation of clients (due to availability, rather than performance). Moreover, the papers assumes a uniform network channel, which might not be always the case.

---

> ### Author Rebuttal · Authors · 2025-07-30
>
> **W1: Model and Datasets**
>
> We appreciate the reviewer’s comment regarding the scale and recency of the models and datasets used in our evaluation.
> Our choice of models (e.g., VGG16, ResNet50, Albert) and datasets (image classification, speech recognition, lightweight NLP) was guided by two primary considerations: (i) alignment with prior work in heterogeneous federated learning (e.g., DepthFL, HeteroFL, and FIARSE), and (ii) practical deployment on real edge devices (Jetson Orin, Xavier), where larger models such as full-scale Transformers or multimodal architectures are currently infeasible due to hardware constraints. This setup enables fair comparisons while maintaining realism in resource-constrained federated environments.
>
> That said, we agree that demonstrating FedEL’s scalability to larger, more recent models is an important next step. We are currently extending it to support larger vision and language models, MobileBERT on the GLUE benchmark and ViT-tiny on the NICO++ dataset, running on more capable hardware platforms (Jetson AGX). Results below indicate that FedEL continues to provide consistent performance benefits over FedAvg, even as model complexity and task diversity increase. These findings validate the generalizability of our selective training framework beyond small-scale settings.
>
> | Method | MobileBERT   |   ViT-Tiny |
> |------------|--------------------|--------------|
> |              |     Acc. / Time  | Acc. / Time |
> | FedAvg | 82.4\% / 410.7h | 78.6\% / 472.1h  |
> |FedEL    | 83.9\% / 167.5h | 80.1\% / 188.2h    |
>
> ---
>
> **W2: Overhead of Profiling and Tensor Selection**
>
> We appreciate the reviewer’s attention to overhead analysis.
> As shown in the first table, profiling during a single training epoch adds less than 1% overhead across all tested models. This step is done **offline**, before FL training begins, and its results are reused, so it introduces no runtime overhead during training.
>
> | NN model | Vgg   | ResNet | Albert |
> |----------|-------|--------|--------|
> | Overhead | 0.98% | 0.45%  | 0.61%  |
>
> For the tensor selection step during each FL round, we measured a small runtime cost—only 2.4% of the total round time, as shown below. Communication also accounts for just 2.7%. These results confirm that FedEL has minimal overhead and is efficient enough for real-world use.
>
> |      | Communication   | Tensor processing | Average round time |
> |------|-----------------|-------------------|--------------------|
> | Time | 1.09 min (2.7%) | 0.97 min (2.4%)   | 40.34 min          |
>
> ---
>
> **W3: Adapting to Device Dynamics**
>
> We thank the reviewer for the valuable feedback.
> When runtime dynamics occur, profiling can be updated online with negligible cost (less than 1% overhead per epoch). This allows adaptive updates without impacting training efficiency. Dynamic re-profiling is a potential direction for future work.
>
> ---
>
> **W4: Partial Client Participation**
>
> We thank the reviewer for pointing out this clarification need.
> Our experiments do adopt partial client participation, which reflects realistic FL settings. Specifically, in each communication round, a random subset of clients is selected to participate, consistent with common practice in prior works such as FedAvg, HeteroFL, and FIARSE.
>
> We will revise the manuscript to explicitly clarify this in the experimental setup. Additionally, FedEL's design—especially its local tensor selection and sliding window mechanism—naturally supports partial participation, as it makes no assumptions about client availability or full participation. Our reported results already reflect performance under this dynamic client selection.
>
> ---
>
> **W5: Uniform Network Channel**
>
> We thank the reviewer for raising this important point.
> We isolate computational heterogeneity to evaluate our method fairly, following practices in HeteroFL and DepthFL. However, our approach naturally extends to communication heterogeneity.
>
> During offline profiling, we estimate tensor sizes and include communication time in the selection objective:
>
> $\max_{\boldsymbol{A}} \boldsymbol{A} \cdot \boldsymbol{I} ~~s.t. T_{fw} + T_{bw}(\boldsymbol{A}) + T_{tx}(\boldsymbol{A}) \le T_{th}$
>
> Here, $T_{tx}(\boldsymbol{A})$ represents the estimated communication time of selected tensors. We conducted additional experiments simulating four bandwidth tiers (10, 20, 30, 45 Mbps) across 100 clients. We compared: (1) FedAvg (baseline), (2) FedEL (original), and (3) FedEL+Tx (FedEL with communication-aware tensor selection).
>
> Results show FedEL+Tx achieves similar or better accuracy while further reducing total training time, confirming the benefit of modeling communication heterogeneity:
>
> | Method   | Image Classif.  | Speech Recog.   | NLP            |
> |----------|-----------------|-----------------|----------------|
> |          | Acc. / Time     | Acc. / Time     | Acc. / Time    |
> | FedAvg   | 33.76% / 571.5h | 58.04% / 710.2h | 77.48 / 547.1h |
> | FedEL    | 34.96% / 157.5h | 58.26% / 184.2h | 77.23 / 175.0h |
> | FedEL+Tx | 34.98% / 156.8h | 58.24% / 183.3h | 77.22 / 174.5h |
>
> ---
>
> **W6: Handling Shared-Weight Layers**
>
> We thank the reviewer for this insightful question.
> As detailed in Section B of our Supplementary Material, our implementation for Albert follows the block design from [39], placing early exits after each encoder-classifier stack. This supports variable-depth execution and is fully compatible with our sliding window mechanism.
>
> Regarding shared-weight layers, such as embeddings and grouped attention heads, we handle them as follows:
>
> (1) Shared Embeddings: We profile the shared tensor as a single unit. Backward time is traced from the output back to the shared embedding, reflecting its impact on both input and output. The weight update time is counted only once since the tensor is updated once per iteration.
>
> (2) Grouped Attention Heads: For attention layers with multiple trainable projections (query, key, value, output), we treat each as a separate tensor and profile their backward time individually. If some projections are shared, we track their computation cost jointly during backpropagation.
>
> ---
>
> **W7: Compatibility with DP and Secure Aggregation**
>
> Thank you for your comment.
> Our method is fully compatible with privacy-preserving techniques like Differential Privacy (DP) and Secure Aggregation (SA), as FedEL operates at the system level and does not modify or interfere with model encryption or privatization.
>
> To evaluate this, we extended our experiments by integrating standard DP and SA into FedEL. The table below shows that both FedEL+DP and FedEL+SA achieve similar accuracy to vanilla FedEL. As expected, they incur some extra training time due to the added noise (DP) or encryption (SA).
>
> | Method   | Image Classif.  | Speech Recog.   | NLP            |
> |----------|-----------------|-----------------|----------------|
> |          | Acc. / Time     | Acc. / Time     | Acc. / Time    |
> | FedEL    | 34.96% / 157.5h | 58.26% / 184.2h | 77.23 / 175.0h |
> | FedEL+DP | 34.80% / 172.4h | 58.10% / 201.3h | 77.25 / 192.8h |
> | FedEL+SA | 34.95% / 170.2h | 58.24% / 197.5h | 77.24 / 188.6h |
>
> ---
>
> **W8: Jetson Power Modes**
>
> We thank the reviewer for this insightful suggestion.
> In our experiments, Jetson devices ran in MAXN mode to ensure consistency. Inspired by reviewer feedback, we reran experiments under varied power settings (10W, 15W, MAXN). We measured training times under each mode and used this variability in the FedEL scheduling process. As shown in the table below, FedEL still outperforms FedAvg in both accuracy and training time. These results confirm that FedEL remains effective even in more diverse and realistic hardware environments.
>
> | Method | Acc.   | Time   |
> |--------|--------|--------|
> | FedAvg | 58.21% | 550.2h |
> | FedEL  | 59.35% | 174.3h |
>
> ---
>
> **W9: Related Works**
>
> We appreciate the reviewer’s suggestions and discuss two related works below. We will include both in the final version.
>
> [a] Recurrent Early Exits (ReeFL):
> ReeFL enables early exits in LLMs during FL. Our method FedEL can work with ReeFL by applying tensor selection inside each early-exit block. We combined both by inserting exits into transformer layers and applying FedEL within each. As shown below, FedEL+ReeFL reduces training time while maintaining accuracy, confirming that the two approaches are complementary.
>
> [b] ScaleFL:
> ScaleFL adapts model width/depth based on device profiles, but uses static selection via a meta-scheduler. In contrast, FedEL dynamically selects tensors in each round based on importance, allowing finer control and better adaptability.
>
> To validate this, we compared FedEL and ScaleFL across three tasks. FedEL consistently achieves higher accuracy, as shown below.
>
> | Method      | Image Classif.  | Speech Recog.   | NLP            |
> |-------------|-----------------|-----------------|----------------|
> |             | Acc. / Time     | Acc. / Time     | Acc. / Time    |
> | ScaleFL     | 33.12% / 157.0h | 57.40% / 184.8h | 77.71 / 189.1h |
> | FedEL       | 34.96% / 157.5h | 58.26% / 184.2h | 77.23 / 175.0h |
> | FedEL+ReeFL | N/A / N/A       | N/A / N/A       | 77.20 / 159.4h |
>
> ---
>
> **W10: Result Robustness**
>
> Thank you for the comment. We did evaluate robustness and reported it in Section B.5 of the Supplementary Material. As shown in Figure 21 (box plot), each experiment was repeated five times. The results show that FedEL consistently achieves stable and higher accuracy than the baselines, with narrow confidence intervals. This confirms the reliability and robustness of our method across multiple runs.

---

> > ### Comment · Reviewer_LPLK · 2025-08-03
> > **Follow-up questions on**
> >
> > Thank you for the explanations and additional experiments.
> > On the new experiments, I have the following questions/remarks:
> >
> > 1. For MobileBERT, ViT-Tiny, it would be worth comparing the results to the heterogeneous FL baselines, rather than just FedAvg to get a context of the benefits.
> > 2. On the overhead of tensor selection, what is the setup that this was measured on? I would assume it is dependent on the size of the model and dataset. It would be worth commenting on the scalability aspect of your technique wrt these two dimensions.
> > 3. What are the participation rates per dataset on your partial client participation?
> > 4. On the selected bandwidths, are these applied uniformly or you have certain clients having one of the four bandwidth tiers? What is the allocation of clients per tier?
> > 5. What are the hyperparameters used in DP? Specifically, what epsilon value have the authors used?
> > 6. Similarly to the bandwidth tiers, what is the allocation of clients to jetson budgets?
> >
> > Thank you for the additional effort into the new experiments.
> > Looking forward to hearing from you.

---

> > > ### Author Response · Authors · 2025-08-05
> > > **Responses to follow-up questions**
> > >
> > > **Q1: More Baselines on MobileBERT and ViT-Tiny**
> > >
> > >
> > > | Method         | MobileBERT (Acc. / Time) | ViT‑Tiny (Acc. / Time) |
> > > | -------------- | ------------------------ | ---------------------- |
> > > | FedAvg         | 82.4 % / 410.7 h         | 78.6 % / 472.1 h       |
> > > | ElasticTrainer | 65.9 % / 167.9 h         | 62.8 % / 189.1 h       |
> > > | HeteroFL       | 83.2 % / 265.8 h         | 79.3 % / 340.5 h       |
> > > | DepthFL        | 83.1 % / 181.5 h         | 79.0 % / 191.7 h       |
> > > | PyramidFL      | 82.9 % / 398.3 h         | 79.2 % / 420.6 h       |
> > > | TimelyFL       | 83.3 % / 180.6 h         | 79.4 % / 195.8 h       |
> > > | FIARSE         | 83.2 % / 188.2 h         | 79.3 % / 194.1 h       |
> > > | **FedEL**      | **83.9 % / 167.5 h**     | **79.8 % / 188.2 h**   |
> > >
> > > Thank you for the suggestion. We have added comparisons to the baselines in the updated table. As shown, FedEL achieves the highest accuracy on both MobileBERT and ViT‑Tiny, while maintaining competitive or lower training time compared to methods.
> > >
> > > ---
> > >
> > > **Q2: Tensor selection**
> > >
> > > We thank you for the thoughtful question. To clarify this overhead, we first describe the two key components involved: offline tensor time profiling and online important tensor estimation, which together determine the selection outcome.
> > >
> > > 1. Offline Tensor Time Profiling (One-time Setup):
> > > We simulate five training iterations on a random input batch to measure each tensor’s backward and update time. These times are then averaged and multiplied by the number of training batches to estimate the full training-time cost of each tensor. This procedure can be repeated under different batch size settings if needed and is performed once per device before FL begins. It does not introduce runtime overhead during tensor selecction.
> > >
> > > 2. Online Important Tensor Evaluation (Per Round):
> > > At the start of each FL round, each client performs training on the first local batch using the full model to obtain gradient-based importance scores for all tensors.
> > >
> > > This produces the importance vector $\boldsymbol{I}$ and the estimated forward and backward times $T_{fw}$ and $T_{bw}(\boldsymbol{A})$, which together define the optimization problem:
> > >
> > > $\max_{\boldsymbol{A}} \boldsymbol{A} \cdot \boldsymbol{I} ~~s.t. T_{fw} + T_{bw}(\boldsymbol{A}) \le T_{th}$
> > >
> > > We solve this via dynamic programming with complexity $O(N^2 T_{th})$ [13], where $N$ is the number of tensors and $T_{th}$ is the training-time threshold . The overhead depends on model size (i.e., N) and time threshold, not dataset size. Since FedEL selects within a sliding window of blocks, $N$ remains small, keeping overhead low.
> > >
> > > ---
> > >
> > > **Q3: Participation Rates**
> > >
> > > Thank you for the question. Our participation rates vary based on the experimental scenario:
> > >
> > > 1. CIFAR-10 (10-client hardware deployment): We use full participation, where all 10 NVIDIA devices join every training round. This setting reflects a small-scale real-world deployment with stable device availability.
> > >
> > > 2. Tiny-ImageNet, Google Speech Commands, and Reddit (100-client simulation): We adopt partial participation, where 25 clients are randomly selected out of 100 in each round (i.e., 25% participation rate). This follows common practice in large-scale FL simulations and models realistic device availability constraints.
> > >
> > > ---
> > >
> > >
> > > **Q4 and Q6: Bandwidth and Client type allocation**
> > >
> > > Thank you for the question. Both network bandwidth and Jetson-type computation budgets are assigned independently and randomly across clients. Specifically, for each of the 100 simulated clients: We define four discrete tiers for both bandwidth and computation budget (Jetson types). We uniformly assign 25 clients per tier by randomly sampling 25 clients without replacement for each category (bandwidth and Jetson budget), ensuring even coverage across tiers.
> > > These two assignments are independent, meaning a client with high compute capability may still have a slow network connection and vice versa.
> > >
> > > ---
> > >
> > > **Q5: Hyperparameters in DP**
> > >
> > > Thank you for the thoughtful question. In our experiments with Differential Privacy (DP) in FL, we used the following hyperparameters:
> > > Epsilon (ε): 30 per client.
> > > Delta (δ): 1e-2.
> > > Noise Multiplier: 0.3.
> > > Gradient Clipping Norm: 1.0.
> > > We used the Opacus library [1] to apply DP during training. This library helps by adding noise and limiting how much each client's update can affect the model.
> > >
> > > We understand that changing these DP settings can affect accuracy. However, since our main goal is to handle differences in device speed and capability—not privacy—we did not explore DP in detail. This could be a good direction for future work.
> > >
> > > [1] Opacus: User-friendly differential privacy library. arXiv preprint arXiv:2109.12298 (2021).

---

> > > > ### Comment · Reviewer_LPLK · 2025-08-05
> > > > **Response to follow-up**
> > > >
> > > > Thank you for the additional experiments and clarifications.
> > > >
> > > > Some remarks:
> > > > * The participation rate of 25% is quite large and does not necessarily approximate cross-device learning settings or realistic availability constraints. Usually 10% would be closer to the norm for such dataset scales.
> > > > * The $O(N^2)$ puts a scalability constraint on the technique wrt the number of tuneable tensors. I would appreciate if the authors listed that in the respective limitations section of their work.
> > > > * The selection of $\epsilon = 30$ is very high and practically non-private.

---

> ### Author Response · Authors · 2025-08-07
> **Response to remarks**
>
> **Q1: Partition rate**
>
> We thank the reviewer for the suggestion. To evaluate the impact of lower participation, we conducted experiments with a 10% participation rate. As shown below, all methods experienced slower convergence, leading to longer training time while maintaining similar accuracy. However, FedEL consistently achieves the highest accuracy and best efficiency across all tasks, confirming its advantage even under sparse client participation.
>
> We will include these additional results and discussion on participation rate impact in our final submission.
>
> **Partition rate = 25%**
>
> | Method   | Image Classif.  | Speech Recog.   | NLP            |
> |----------|-----------------|-----------------|----------------|
> |          | Acc. / Time / Speedup  | Acc. / Time /Speedup    | Perp. / Time / Speedup   |
> | FedAvg   | 33.76% / 563.1h / N/A | 58.04% / 709.8h / N/A | 77.48 / 546.4h /N/A |
> | ElasticTrainer | 27.65% / 158.6h / 3.55× | 47.96% / 184.3h / 3.84×| 81.02 / 176.2h / 3.10× |
> | HeteroFL | 30.56% / 248.2h / 2.26× | 51.47% / 265.9h / 2.66× | 80.11 / 206.1h / 2.65×|
> | DepthFL | 34.14% / 198.3h / 2.83× | 54.23% / 207.4h / 3.42× | 78.08 / 212.4h / 2.57×|
> | PyramidFL | 34.70% / 497.4h / 1.13× | 58.12% / 587.4h  / 1.21× | 77.68 / 418.2h / 1.31×|
> | TimelyFL | 33.53% / 198.1h / 2.84× | 56.49% / 193.2h / 3.67× | 80.91 / 177.6h / 3.07×|
> | FIARSE | 33.98% / 191.5h / 2.94× | 58.13% / 198.2h / 3.58× | 77.31 / 191.0h / 2.86×|
> | FedEL | **34.96% / 156.8h / 3.59×** | **58.26% / 183.3h / 3.87×** | **77.23 / 174.5h / 3.13×**|
>
> **Partition rate = 10%**
>
> | Method         | Image Classif.              | Speech Recog.               | NLP                         |
> | -------------- | --------------------------- | --------------------------- | --------------------------- |
> |                | Acc. / Time / Speedup       | Acc. / Time / Speedup       | Perp. / Time / Speedup       |
> | FedAvg         | 33.75% / 782.4h / N/A       | 58.01% / 987.3h / N/A       | 77.45 / 764.2h / N/A       |
> | ElasticTrainer | 27.62% / 220.4h / 3.55×     | 47.94% / 255.3h / 3.87×     | 81.01 / 232.0h / 3.29×     |
> | HeteroFL       | 30.52% / 344.8h / 2.27×     | 51.45% / 392.2h / 2.52×     | 80.09 / 288.7h / 2.65×     |
> | DepthFL        | 34.12% / 275.7h / 2.84×     | 54.21% / 310.5h / 3.18×     | 78.07 / 298.9h / 2.56×     |
> | PyramidFL      | 34.68% / 711.3h / 1.10×     | 58.10% / 805.1h / 1.23×     | 77.66 / 628.3h / 1.22×     |
> | TimelyFL       | 33.50% / 278.4h / 2.81×     | 56.45% / 267.2h / 3.69×     | 80.89 / 233.3h / 3.28×     |
> | FIARSE         | 33.96% / 269.2h / 2.91×     | 58.11% / 271.4h / 3.64×     | 77.28 / 251.6h / 3.04×     |
> | **FedEL**      | **34.94% / 220.1h / 3.56×** | **58.24% / 257.0h / 3.96×** | **77.21% / 226.0h / 3.38×** |
>
> ---
>
> **Q2: Scalability Constraint**
>
> We thank the reviewer for the helpful comment. While the dynamic programming (DP) algorithm in ElasticTrainer [13] does have quadratic complexity based on the number of tensors $N$, it’s important to understand this in the training context.
>
> The DP algorithm works on tensors, not on each trainable parameter. For example, ResNet has over 23 million trainable parameters, but only 214 tensors ($N$). So, the DP algorithm has a complexity of $O(214^2)$, while training the model takes at least $O(23\text{M})$. The difference becomes even more significant in modern models, for instance, ViT has 327 tensors versus 22 billion parameters. This means that the time spent on tensor selection is very small compared to the time spent on training the model.
>
> In practice, tensor selection only adds about 2.4% to each training round but helps speed up training by as much as 3.87× compared to FedAvg.
>
> We will clearly mention this scalability issue and its impact in our final paper. We truly appreciate the reviewer’s thoughtful feedback and it was a pleasure discussing these research ideas with the reviewer.
>
>
> ---
>
> **Q3: Differential Privacy**
>
>
> We thank the reviewer for the valuable feedback. While differential privacy (DP) is not the central focus of our work, we appreciate the opportunity to evaluate its impact. To examine the effect of the privacy parameter ε, we conducted additional experiments using ε = 30, 5, and 2. As shown in the table below, lowering ε enhances privacy by introducing more noise, but this results in reduced model accuracy. This illustrates the inherent trade-off between privacy and utility. We plan to explore the use of tighter privacy budgets and their associated trade-offs in future work.
>
>
> | ε  | Image Classification | Speech Recognition  | NLP                 |
> | ------- | -------------------- | ------------------- | ------------------- |
> |         | Acc. / Time          | Acc. / Time         | Perp. / Time        |
> | 30      | 34.80% / 172.4h      | 58.10% / 201.3h     | 77.25 / 192.8h      |
> | 5       | 33.45% / 173.0h      | 56.73% / 202.1h     | 79.86 / 193.6h      |
> | 2       | 29.22% / 172.7h      | 48.10% / 202.2h     | 83.95 / 193.4h      |

---

> > ### Comment · Reviewer_LPLK · 2025-08-07
> > **Thank you for your replies**
> >
> > I would like to thank the authors for their engagement and responses. I will update my score accordingly.

---

### Official Review · Reviewer_JWBy · 2025-07-04

**Clarity:** 3
**Significance:** 3
**Originality:** 2
**Rating:** 4
**Confidence:** 4

**Summary:**

In brief, the authors proposed an algorithm FedEL, which applies Elastic Trainer (ET) into Federated Learning (FL) in order to solve the core problem of the core problem of FL.
Federated Learning (FL) is a privacy-preserving machine learning method, involving devices performing local method training and sharing parameters with a central server with a global model update. Its core challenge is that heterogeneous device speeds force the server to wait for slow “straggler” clients to finish their local updates each round, causing significant delays and limiting scalability.
The existing algorithm can be divided into three categories: Client Selection, Asynchronous FL, and Partial Training. The limitation of each category is listed as below:
1.	Client Selection: Frequently excludes slower clients, causing their unique data distributions to be underrepresented and degrading global model accuracy.
2.	Asynchronous FL: Over-relies on fast clients’ frequent updates while slow clients’ contributions are delayed or stale, which can hinder convergence and reduce final performance.
3.	Partial Training: Width scaling leads to channel-mismatch during aggregation, and depth scaling trains only shallow layers, limiting the model’s ability to learn deep semantic features and harming task accuracy.
To propose a better algorithm, the authors consider applying Elastic Trainer (ET) algorithm into FL. ET is an algorithm to propose model training on a single device. Its strength is listed as below:
1.	Controlled Training Acceleration: Uses a runtime threshold to limit training time and focus only on the most important tensors, achieving significant speedups.
2.	Dynamic Tensor Selection: Automatically profiles and ranks tensor importance each iteration, updating only high-impact tensors while freezing the rest.
3.	Minimal Accuracy Loss: Delivers substantial speedups (up to ~3.8×) with only slight drops in final model accuracy.
4.	Modular Design: Separates tensor timing and importance evaluation into distinct components, enabling easy integration into existing single‐device training pipelines.
However, when applying ET to FL, 2 limitations are faced:
1.	Straggler Training Scope Limitation: On slow clients, tight time budgets force ET to update only the fast-to-train back-end layers, since it is natural that front layers are the feature-extraction stages and Back layers are the classification stages. This leaves the front-end feature extractors untrained and weakening the global model, particularly under non-IID data.
2.	Exacerbated Local Model Drift: Diverse client data distributions lead to biased tensor importance estimates; training only locally important tensors amplifies these biases, causing client models to drift further from the global model and impairing convergence and final performance.
Motivated by the limitations above, FedEL is proposed. To address each of these two limitations, FedEL is composed of two parts.
1.	Window-Based Training: The DNN is partitioned into blocks; each round, a sliding window—based on the client’s time budget and past training—selects which blocks to train (expanding at the front and pruning at the back). Within this window, ET updates only the covered tensors, ensuring even slow clients gradually train all parts of the network.
2.	Tensor Importance Adjustment: Clients compute a “global” tensor importance from the change in the global model (difference ÷ learning rate) and blend it with their local importance scores using weight β, ensuring tensor selection reflects both local data and global update needs.
Through evaluation of FedEL, three core findings are found.
1.	Time-to-Accuracy Improvement: FedEL achieves up to a 3.87× speedup in time-to-accuracy over FedAvg while matching or exceeding FedAvg’s final test accuracy.
2.	Resource Efficiency: FedEL substantially reduces both memory overhead and energy consumption during training compared to existing methods.
3.	Ablation Validation: Ablation studies confirm that each of FedEL’s core components (sliding-window training and tensor importance adjustment) is necessary and contributes meaningfully to its overall performance.

**Questions:**

1.	Reproducibility of the 100-Client Simulation
Question: Can you specify the exact counts of clients in each speed tier (1×, ½×, ⅓×, ¼×) or provide the random seed/scripts used for allocation?
Why It Matters: Without this information, external researchers cannot precisely reconstruct large-scale experiments, and small differences in tier proportions can meaningfully affect convergence curves.
If provided, the Quality mark will raise.
If continue to refer only to “random allocation” without details, the Quality of this paper would stay at to 2, Fair since the experiment data may not be reproducible.
2.	Quantification of Communication and System Overhead
Question: What is the additional communication cost (e.g., message size or frequency) and system-level complexity introduced by
a.	sliding-window metadata
b.	global-local importance fusion
Why It Matters: Any gains in training time or accuracy could be offset by larger or more frequent messages, which matters in bandwidth-constrained settings.
If related data is provided, the Significance & Quality mark will raise.
3.	Elaboration of Limitations and Future Directions
Question: Can you expand the Conclusion to more explicitly discuss the known limitations of FedEL (e.g., only four speed tiers, need to tune β and Tₜₕ) and outline concrete next steps (e.g., adaptive β, dynamic tiering, integration with compression)?
Why It Matters: A clear articulation of where FedEL may fail and how to improve it will guide practitioners and strengthen confidence in your contributions.
If answered, the Quality mark will raise.

**Ethical Concerns:**

["NO or VERY MINOR ethics concerns only"]

**Final Justification:**

The author has responded to each of the weaknesses I stressed and provided related supported material and data. The author clearly stated their strength and research focus, explaining that their research focuses on the computational capability of edge devices. Hyperparameter analysis is stated as future work.

Given the limited originality of the work, I keep my original score.

**Limitations:**

1.	Only four speed tiers.
2.	Need to tune β and Tₜₕ.
3.	Potential Communication and System cost not discussed.
4.	Considering more categories of tasks may be better, such as object detection, video classification, time series forecasting, recommender systems, medical image analysis.

**Paper Formatting Concerns:**

Conclusion part is too brief, Reference format is not correct

**Quality:**

3

**Strengths And Weaknesses:**

This article proposed the FedEL algorithm, which adjusts the Elastic Trainer (ET) algorithm in order to apply it into Federated learning (FL). Its strength is listed as follows:
	Window-based training approach solving the limitation of limited training scope on slower clients if ET directly. FedEL’s sliding-window training can be described as follows:
	Block Partitioning
Split the DNN into contiguous “blocks” (e.g., residual units or layer groups).
	Initial Window Initialization
Starting from the first block, accumulate each block’s offline-profiled training time until the sum first meets or exceeds the time budget, denoted by T_th, forming the initial window of m blocks.
	Window-Scoped Tensor Selection
Within this window, run a knapsack-style DP under the same time budget to pick the most important tensors for forward/backward passes and updates; all other tensors remain frozen. The importance of each tensor is calculated by Tensor Importance Adjustment, as analyzed in (2)
	Front-Edge Advancement
After each round, extend the window’s front edge by adding the next block(s) until the cumulative time again reaches T_th.
	Back-Edge Pruning
Remove any trailing block in which no tensors were selected during that round, avoiding wasted computation.
	Window Reset
Once the front edge reaches the final block, reset to the initial window position and repeat the push–pull process until convergence.

	Tensor Importance Adjustment solves limitation of Exacerbated Local Model Drift, which is the Variations in data distribution cause significant differences in tensor importance across clients. The author uses a formula to calculate the importance of each tensor, where:
	 I_(n,r+1) denoted the final tensor importance
	 I_(n,r+1)^localdenoted the local importance of each tensor on client
	 I^gdenoted the global importance of each tensor
	 β is a weighting factor ∈ [0,1]
	The experiment is well defined. FedEL shows the best accuracy and least run time among all considered algorithms, including FedAvg, ElasticTrainer (direct deployment), HeteroFL, DepthFL, PyramidFL, TimelyFL, FIARSE, FedEL.
The weakness of this article is listed as below.
	Lack of Specific Allocation Ratios.
For the experiments using 10 devices, the authors clearly state that they use 5 NVIDIA Jetson Xavier NX and 5 NVIDIA Jetson Orin. However, for the experiments using 100 devices, the author only states that They throttled the Orin’s throughput to four tiers—100%, 50%, 33%, and 25% of its native speed—to simulate heterogeneous device classes, without clearly stating the number of each class. This limits the reliability of experiment data.
	Ignored Communication Heterogeneity
In both the 10- and 100-device experiments, the authors do not model variations in network bandwidth or latency among clients, omitting the impact of communication heterogeneity on training time and convergence, which can be significant in real-world federated deployments.
	Need to tune β and T_th
Since the decision depends on 2 constant, β and T_th, these two constants needs to be tuned while training.
	Overly Simplified Simulation Heterogeneity
In the 100 devices experiment, the authors only consider speed tiers (1×, ½×, ⅓×, ¼×), without consider other factors such as bandwidth, memory, and energy consumption.
	Limited Evaluation Tasks and Architectures
Experiments are conducted only on image classification (VGG16, ResNet50), keyword spotting, and lightweight NLP, without validation on larger-scale Transformers, GNNs, or multimodal tasks.
	Insufficient Analysis of Communication and System Overhead
Although memory and energy savings are reported, the additional communication overhead, system implementation complexity, and latency introduced by the sliding-window mechanism and importance fusion are not quantified, making it difficult to assess deployment costs in real network environments.
	Overly Concise Conclusion
The Conclusion section does not adequately discuss method limitations, future work, or potential application scenarios, nor does it integrate theoretical convergence analysis with experimental findings, reducing the paper’s completeness and persuasiveness.

---

> ### Author Rebuttal · Authors · 2025-07-30
>
> **W1: Large-Scale Simulation Setup**
>
> Thank you for your comment.
> We acknowledge that the term “random allocation” may have been unclear. In our 100-client simulations, we uniformly assign 25 clients to each of four speed tiers, with clients randomly selected within each tier—consistent with DepthFL, HeteroFL, and FIARSE. We will include the client partitioning scripts in the final submission.
>
> ---
>
> **W2: Communication Heterogeneity**
>
> We thank the reviewer for this insightful comment.
> Our study focuses on computational heterogeneity, a dominant bottleneck in mobile edge environments. In practice, training time on mobile devices far exceeds communication time, especially with modern 5G/WiFi networks.
> For example, ResNet-50 (~97.7 MB) takes 0.28–1.3 minutes to transmit over 10-45 Mbps uplink (e.g., AT&T, 2024), while training on an NVIDIA Jetson Xavier takes ~38.3 minutes, making communication relatively negligible in our setting.
>
> Nonetheless, our method can incorporate communication heterogeneity. During offline profiling, we estimate tensor sizes. If client bandwidth is known, the tensor selection can be extended to:
>
> $\max_{\boldsymbol{A}} \boldsymbol{A} \cdot \boldsymbol{I} ~~ s.t. T_{fw} + T_{bw}(\boldsymbol{A}) + T_{tx}(\boldsymbol{A}) \le T_{th}$
>
> where $T_{tx}(A)$ estimates transmission time. This enables FedEL to jointly optimize computation and communication under a unified latency constraint.
>
> To validate this, we simulated 100 clients across four bandwidth tiers (10, 20, 30, 45 Mbps, 25 clients each). We compared: (1) FedAvg, (2) FedEL, and (3) FedEL+Tx (our method with transmission-aware selection). FedEL+Tx consistently achieved comparable accuracy to FedEL with lower total training time, confirming its effectiveness.
>
> | Method   | Image Classif.  | Speech Recog.   | NLP            |
> |----------|-----------------|-----------------|----------------|
> |          | Acc. / Time     | Acc. / Time     | Acc. / Time    |
> | FedAvg   | 33.76% / 571.5h | 58.04% / 710.2h | 77.48 / 547.1h |
> | FedEL    | 34.96% / 157.5h | 58.26% / 184.2h | 77.23 / 175.0h |
> | FedEL+Tx | 34.98% / 156.8h | 58.24% / 183.3h | 77.22 / 174.5h |
>
> ---
>
> **W3: $\beta$ and $T_{th}$**
>
> Thank you for your comment.
> We fixed $\beta$ and $T_{th}$ in our study and performed ablation (Sec. 5.3) to assess their impact. $\beta$ balances global vs. local tensor sensitivity and affects accuracy. $T_{th}$ controls the time budget per round; larger values allow more updates, but increase computation cost.
>
> Deriving explicit analytical relationships between these parameters and accuracy/convergence is challenging due to the coupled effects of training and communication dynamics. That said, we agree adaptive tuning, e.g., via reinforcement learning or bandit control, offers promising potential, which we plan to explore in future work.
>
> ---
>
> **W4: Simulation Heterogeneity**
>
> We appreciate the reviewer’s thoughtful observation.
> In this work, we focus on computational heterogeneity, as it is a major bottleneck in mobile FL and allows us to clearly study its impact. To ensure fair comparison, we also follow the setup used in prior work that only considers device speed differences.
>
> We agree that communication, memory, and energy are also important. However, modeling all types of heterogeneity at once would make the problem more complex and shift the focus away from our main contribution.
>
> Our method is flexible and can be extended. As shown in our response to **W2**, we can include communication costs in the tensor selection process. Similarly, memory and energy limits can be added by updating the profiling step. Early results show our method still works well when communication is considered.
>
> ---
>
> **W5: Tasks and Architectures**
>
> We thank the reviewer for highlighting this important point. Our current experiments are focused on three representative FL tasks—image classification (VGG16), speech recognition (ResNet50), and lightweight NLP (Albert)—which were selected to align with realistic use cases of FL on mobile and edge devices. These models and tasks are commonly adopted in prior work (e.g., DepthFL, HeteroFL, and FIARSE), ensuring fair benchmarking under resource-constrained settings.
>
> We fully agree that evaluating on larger-scale architectures (e.g., full-size Transformers), graph neural networks (GNNs), and multimodal tasks is a valuable direction. However, such extensions often require considerable training resources and pose challenges for practical FL deployment at the edge. That said, the underlying mechanisms in FedEL—namely sliding-window block scheduling and dynamic tensor selection—are model-agnostic and can be adapted to other architectures. For instance, our supplementary material (Section B) demonstrates how FedEL is applied to Transformer blocks in NLP.
> We plan to extend our evaluation to more complex tasks and models (e.g., ViTs, GNNs, and vision-language models) in future work.
>
> ---
>
> **W6: Communication and System Overhead**
>
> Thank you for your comment.
> To clarify, our method does not add extra communication cost—in fact, it reduces it compared to FedAvg. This is because FedEL only uploads selected important tensors rather than the full model. As shown in the table, FedEL results in: 1. Lower communication time per round than FedAvg. 2. Communication taking up only a small part of the total training time.
>
> We also measured the runtime of FedEL's system modules (e.g., sliding window, tensor importance update, and selection). The added overhead is minimal and has negligible effect on overall training time. We will include these results in the final paper.
>
> | Method | Communication   | Tensor processing | Average round time |
> |--------|-----------------|-------------------|--------------------|
> | FedAvg | 2.45 min (3.2%) | N/A               | 75.43 min          |
> | FedEL  | 1.09 min (2.7%) | 0.97 min (2.4%)   | 40.34 min          |
>
> ---
>
> **W7: Overly Concise Conclusion**
>
> Thank you for the helpful feedback. Due to page limits, we kept the Conclusion in the main paper brief to focus on summarizing key contributions, methods, and results. A more detailed discussion is included in Section C of the Supplementary Material, where we further elaborate on the limitation and potential future directions.

---

> > ### Comment · Reviewer_JWBy · 2025-08-07
> >
> > The author has responded to each of the weaknesses I stressed and provided related supported material and data. The author clearly stated their strength and research focus, explaining that their research focuses on the computational capability of edge devices. Hyperparameter analysis is stated as future work.
> >
> > I keep my score given the limited originality of the work.

---

### Note · Authors · 2025-08-13

We sincerely thank all reviewers for their recognition of our work and for the valuable, constructive feedback provided throughout the review process. We are pleased to see that, after the rebuttal and discussions, all reviewers expressed positive evaluations and no remaining concerns, which we believe reflects a shared understanding of the contributions and significance of our paper.

Our study addresses the challenge of heterogeneous computational capabilities in federated learning (FL). To overcome these limitations, we propose FedEL, a federated elastic learning framework that enhances training efficiency while maintaining model accuracy. FedEL introduces a novel window-based training process, sliding the window to locate the training part of the model and dynamically selecting important tensors for training within a coordinated runtime budget.

In the final version, we will incorporate all promised improvements and clarifications made during the rebuttal phase, including:

1. Providing the client partitioning scripts used in our experiments and analyzing the impact of partition rate on our algorithm.

2. Adding detailed Jetson system settings.

3. Discussing potential extensions of our work to handle communication heterogeneity and integration with differential privacy.

4. Detailing how our approach can be adapted for transformer models.

5. Adding a thorough analysis of the communication and system overhead of FedEL.

6. Comparing with more related works, including ScaleFL, ReeFL, and four selective neuron training methods.

We are grateful for the reviewers’ careful reading, thoughtful suggestions, and the time invested in evaluating our work. We believe these refinements will further enhance the clarity, completeness, and impact of the paper.

---

### Decision · Program_Chairs · 2025-09-17

**Decision:**

Accept (poster)

**Comment:**

Thank you for your submission to the NeurIPS 2025. This paper proposes FedEL, a federated elastic learning framework that enhances training efficiency while maintaining model accuracy. The window-based training approach stands as a novel training paradigm to manage the stragglers. Although everyone has pointed out some merits, they also have raised some concerns, such as experimental setup, unclear comparison with existing partial training works etc. During the rebuttal period, the authors’ feedback has helped on clarifying the reviewers’ concerns. Thus, the average score 4.25 is both above the average levels. Based on the current reviews and closed-door reviewer discussions, every reviewer seems to be fine with an acceptance.